# Unraveling CRP/cAMP-mediated metabolic regulation in *Escherichia coli* persister cells

**Han G Ngo[1], Sayed Golam Mohiuddin[1], Aina Ananda[2], Mehmet Orman[1]***

[1]Department of Chemical and Biomolecular Engineering, University of Houston, Houston, United States; [2]Department of Biology, Monmouth University, West Long Branch, United States

## eLife Assessment

The study reports an **important** finding on the role of the global metabolic regulator Crp/cAMP in the formation of antibiotic persister *Escherichia coli*. The evidence supporting the claims is **solid** including metabolomic analysis and characterization of many mutant strains.

**Abstract** A substantial gap persists in our comprehension of how bacterial metabolism undergoes rewiring during the transition to a persistent state. Also, it remains unclear which metabolic mechanisms become indispensable for persister cell survival. To address these questions, we directed our efforts towards persister cells in *Escherichia coli* that emerge during the late stationary phase. These cells have been recognized for their exceptional resilience and are commonly believed to be in a dormant state. Our results indicate that the global metabolic regulator Crp/cAMP redirects the metabolism of these antibiotic-tolerant cells from anabolism to oxidative phosphorylation. Although our data demonstrates that persisters exhibit a reduced metabolic rate compared to rapidly growing exponential-phase cells, their survival still relies on energy metabolism. Extensive genomic-level analyses of metabolomics, proteomics, and single-gene deletions consistently highlight the critical role of energy metabolism, specifically the tricarboxylic acid (TCA) cycle, electron transport chain (ETC), and ATP synthase, in sustaining persister levels within cell populations. Altogether, this study provides much-needed clarification regarding the role of energy metabolism in antibiotic tolerance and highlights the importance of using a multipronged approach at the genomic level to obtain a broader picture of the metabolic state of persister cells.

## Introduction

Bacterial persisters within cell cultures constitute a small subpopulation of cells exhibiting a transient antibiotic-tolerant state (*Balaban et al., 2004*). While persisters have traditionally been characterized as non-growing and dormant phenotypes (*Lewis, 2010*), recent studies challenge these conventional hallmarks, revealing the heterogeneity of persister cells in terms of growth, metabolism, and other cellular activities (*Adams et al., 2011*; *Arnoldini et al., 2014*; *Hossain et al., 2023*; *Ma et al., 2010*; *Mok et al., 2015b*; *Orman and Brynildsen, 2013a*; *Orman and Brynildsen, 2015*; *Shan et al., 2015*; *Wakamoto et al., 2013*). Despite potential discrepancies in research outcomes in the field, we think that these variations arise from the intricate and diverse survival mechanisms employed by bacterial cells in response to adverse conditions, such as antibiotic treatments. Furthermore, the interplay of stochastic and deterministic factors associated with these mechanisms adds another layer of complexity, with outcomes highly contingent on factors such as cell types, antibiotics, and

***For correspondence:**
morman@central.uh.edu

**Competing interest:** The authors declare that no competing interests exist.

experimental and growth conditions (*Michiels et al., 2016*). The persistence phenomenon presents a significant health concern (*Huemer et al., 2020*; *Murray et al., 2022*), as the transient antibiotic-tolerant state of persister cells promotes recurrent infections (*Fisher et al., 2017*) and establishes them as a reservoir for the emergence of antibiotic-resistant mutants (*Barrett et al., 2019*; *Levin-Reisman et al., 2017*; *Windels et al., 2019*).

Drug tolerance is a widespread phenomenon observed in both prokaryotic and eukaryotic cell types. Otto Warburg's research in the early 20th century unveiled an intriguing aspect of mammalian cell metabolism known as 'aerobic glycolysis', wherein proliferating cells (e.g. tumor cells) derive energy predominantly through glycolysis, even in the presence of oxygen (*Warburg, 1925*). This metabolic reprogramming involves restricting entry into the tricarboxylic acid (TCA) cycle through precise enzymatic control, diverting glycolytic intermediates towards anabolic pathways. This adaptation supports the extensive biosynthesis required for active cell proliferation in tumors (*Hanahan and Weinberg, 2011*). Remarkably, tumorigenic persisters that exist in a non-proliferating state may not primarily depend on aerobic glycolysis. Instead, there is substantial evidence suggesting that these cells rely on energy metabolism (*Porporato et al., 2018*; *Vasan et al., 2020*); however, the presence of this metabolic state in antibiotic-tolerant bacteria is still a question mark (*Orman and Brynildsen, 2015*). Reprogramming energy metabolism seems to be an evolutionarily conserved strategy for cells facing stress or adverse conditions, as these cells might benefit from the significantly higher ATP production efficiency provided by oxidative phosphorylation (*Hanahan and Weinberg, 2011*). The identification of a mechanism shared by diverse cell types could open avenues for the development of global strategies to target drug-tolerant cells effectively.

A recent study suggests that bacterial persisters constitute a stochastically formed subpopulation of low-energy cells, despite some observed overlap in ATP levels between antibiotic-sensitive and persister cells (*Manuse et al., 2021*). The persister cells examined in that study were derived from an aged stationary phase culture (48 hr post-inoculation; *Manuse et al., 2021*), a condition known to elevate the number of non-growing cells, which do not promptly resume growth upon transfer to a fresh medium (*Mohiuddin et al., 2020b*). While the non-growing cells formed during the stationary phase have reduced metabolic activity compared to growing cells, as demonstrated in our earlier study (*Orman and Brynildsen, 2013a*), they still exhibit a certain degree of respiratory activity (*Orman and Brynildsen, 2015*). In a recent study measuring ATP levels in viable but non-culturable (VBNC; a phenotype that is antibiotic tolerant but unable to resume growth after antibiotic removal), persister and antibiotic-sensitive cells from 24 hr stationary phase cultures, a significant overlap in intracellular ATP concentrations was observed between antibiotic-sensitive and persister cells (*Li et al., 2024*). On the other hand, VBNC cells exhibited drastically lower ATP levels compared to both persister and antibiotic-sensitive cells (*Li et al., 2024*). Another independent study, which utilized single-cell analysis, reported that ofloxacin persisters were metabolically active cells in exponentially growing cultures before treatment, and these cultures were obtained from 16 hr overnight precultures (not aged; *Goormaghtigh and Van Melderen, 2019*). These findings suggest that while persisters formed during exponential growth may initially retain metabolic activity, they may gradually transition to reduced energy states as they age within the culture. The nature of persister-cell metabolism in bacteria is a topic that has long been a point of contention in scientific circles. The controversy surrounding this topic primarily arises from studies that rely on bacteriostatic chemicals or a limited number of gene deletions, or direct comparisons to exponentially growing cells, all of which have inherent drawbacks (*Conlon et al., 2016*; *Manuse et al., 2021*; *Orman and Brynildsen, 2013a*; *Orman and Brynildsen, 2015*). The metabolism of persister cells, a very complex phenomenon, cannot be easily characterized by a simplistic term such as 'metabolic dormancy'. Even if persister cells may exhibit a lower metabolism compared to the vast majority of rapidly growing exponential-phase cells, they may still rely on energy metabolism for their functioning (*Amato et al., 2014*; *Prax and Bertram, 2014*). In fact, analyzing published studies collectively suggests that bacterial cell metabolism undergoes intricate alterations or rewiring as they transition into a tolerant state, and these alterations seem to be highly dependent on the specific conditions tested (*Adams et al., 2011*; *Arnoldini et al., 2014*; *Ma et al., 2010*; *Mok et al., 2015b*; *Orman and Brynildsen, 2013a*; *Orman and Brynildsen, 2015*; *Shan et al., 2015*; *Wakamoto et al., 2013*).

To gain a better understanding of the critical role of energy metabolism in persister cell survival, we have focused our research on antibiotic-tolerant cells formed during the stationary phase, given that

these cells are known to be highly resilient and capable of surviving a variety of stressors, including antibiotics, and are assumed to be dormant (*Balaban et al., 2004*; *Orman and Brynildsen, 2015*). These cells are also referred to as type I persisters, which cannot readily resume growth when diluted in fresh media during the lag phase (*Balaban et al., 2004*). Previous studies showed that these persister cells can metabolize specific carbon sources that make them susceptible to aminoglycosides (AG) (*Allison et al., 2011*; *Mok et al., 2015a*; *Orman and Brynildsen, 2013b*). Their AG susceptibility is due to increased AG uptake, which is facilitated by increased electron transport chain (ETC) activity and membrane potential (*Allison et al., 2011*). The presence of active energy metabolism in antibiotic-tolerant, non-growing cells may indeed explain their rapid killing by AG in the presence of carbon sources (*Orman and Brynildsen, 2013b*; *Orman and Brynildsen, 2015*). When the knockout strains of global transcriptional regulators (i.e. ArcA, Cra, Crp, DksA, Fnr, Lrp, and RpoS) were screened using the AG potentiation assay in our previous study. The results showed that the panel of carbon sources tested potentiated the AG killing of tolerant cells derived from most knockout strains, except for Δ*crp* and Δ*cyaA* (*Mok et al., 2015a*). This can be attributed to the lack of active energy metabolism in these mutant strains, as the Crp/cAMP potentially shapes persister cell metabolism during the stationary phase. Depletion of primary carbon sources activates adenylate cyclase (CyaA; *Pastan and Perlman, 1970*), increasing cyclic-AMP (cAMP) levels in cells (*Bettenbrock et al., 2007*; *Park et al., 2006*). The cAMP molecules, along with their receptor protein (Crp), activate genes related to the catabolism of secondary carbon sources, potentially supporting cellular functions and energy levels (*Deutscher, 2008*; *Fic et al., 2009*; *Görke and Stülke, 2008*; *Kolb et al., 1993*). Here, using metabolomics, proteomics, and high-throughput screening of single-gene deletion strains, we have provided evidence that the Crp/cAMP regulatory complex maintains an active state of energy metabolism while downregulating anabolic pathways in the antibiotic-tolerant persister cells.

## Results

### Disruption of the Crp/cAMP complex affects the formation of persister cells at the late stationary phase

Since Crp/cAMP-mediated metabolic changes can be induced by nutrient depletion during the stationary phase, we wanted to assess the effects of deleting the *crp* and *cyaA* genes (Δ*crp* and Δ*cyaA*) on both persister cell formation and metabolism during this phase. As anticipated, the deletion of the *cyaA* gene resulted in a notable reduction in intracellular cAMP concentration (*Figure 1—figure supplement 1*). However, the Δ*crp* strain exhibited an increase in cAMP concentration, potentially due to the negative feedback regulatory mechanism of the Crp/cAMP complex for the *cyaA* gene promoter (*Keseler et al., 2005*; *Majerfeld et al., 1981*; *Figure 1—figure supplement 2*). When comparing the growth curves of *E. coli* wild-type (WT), Δ*crp*, and Δ*cyaA* main cultures under identical conditions studied here (see Materials and methods), all three strains started to enter the stationary phase around 5 hr, with the mutant strains exhibiting slightly lower optical density levels than the WT at this time point (*Figure 1—figure supplement 3*). Also, our data provide evidence of an increase in cAMP levels in WT cells during their transition into the stationary phase (*Figure 1—figure supplement 4*), aligning with existing literature (*Bettenbrock et al., 2007*; *Keseler et al., 2005*; *Park et al., 2006*). For type I persister quantification, we diluted cells in the fresh medium, consistent with previous studies (*Balaban et al., 2004*; *Orman and Brynildsen, 2015*), at early (t=5 hr) and late (t=24 hr) stationary phases and subsequently exposed them to an extended period (20 hr) of ampicillin or ofloxacin treatment (200 µg/mL ampicillin and 5 µg/mL ofloxacin). These treatments were carried out at concentrations surpassing the minimum inhibitory concentrations (MIC), necessary for the selection of antibiotic-tolerant persister cells (*Supplementary file 1*; *Keren et al., 2004a*). Also, type I persisters, formed during the stationary phase, exhibit a slow transition from a non-growing state to an active state when transferred to a fresh medium in the lag phase (*Balaban et al., 2004*; *Jõers and Tenson, 2016*; *Orman and Brynildsen, 2015*; *Vulin et al., 2018*); therefore, this transition requires longer antibiotic treatment durations. Moreover, we transferred an equal number of cells from each strain to the fresh medium to ensure consistency in cell numbers. To assess persistence, we collected samples during antibiotic treatment, washed them to remove antibiotics, and plated them on agar media to quantify colony-forming units (CFU) of surviving cells (see Materials and methods). The resulting biphasic kill curves—plots of CFU levels over treatment time—are characteristic of

persistence phenotypes (*Figure 1*). Notably, the WT strain showed a marked increase in both ampicillin- and ofloxacin-persister cells during the late stationary phase, in contrast to the mutant strains, where no such increase was observed (*Figure 1A and B*). However, this trend was not observed in the early stationary phase (*Figure 1A and B*). To confirm that the observed decrease in persister levels in the mutant strains in the late stationary phase is solely attributed to the perturbation of the Crp/cAMP regulatory network, we reintroduced *crp* expression to the Δ*crp* strain using a low-copy plasmid carrying the *crp* gene and its promoter. As a control, we utilized an empty vector of the same plasmid. The results demonstrated that the expression of *crp* restored the persister level in the mutant strain, while the plasmid itself had no impact on persister levels (*Figure 1—figure supplement 5A, B*). Altogether, these findings highlight the significant role of Crp/cAMP in ampicillin and ofloxacin persister formation in the late stationary phase.

Ampicillin and ofloxacin, both broad-spectrum antibiotics with a strong dependence on cell metabolism (*Zheng et al., 2020*), target cell wall synthesis and DNA gyrase activity, respectively. In addition to these two antibiotics, we also quantified AG-persister levels in both WT and mutant strains. This was achieved by exposing diluted cells from both early and late stationary phases to 50 μg/mL gentamicin, a concentration exceeding the MIC levels (*Supplementary file 1*). After prolonged gentamicin exposure (20 h), the tolerant cell colonies were found to be below the limit of detection for all strains and conditions (*Figure 1C*, *Figure 1—figure supplement 5C*). Although bacterial tolerance can vary significantly depending on the specific antibiotics and growth phase used (*Hofsteenge et al., 2013*), this outcome contrasts starkly with the observed levels of ampicillin and ofloxacin persisters in WT (*Figure 1*). The mechanism by which AGs eliminate persister cells in the WT strain may be linked to their metabolism, given that AG uptake is an energy-requiring process (*Allison et al., 2011*; *Taber et al., 1987*), and their metabolism will be explored further in the subsequent section.

We would like to highlight that antibiotic concentrations, washing procedures (to remove antibiotics), and agar plate incubation times for CFU enumeration may affect experimental outcomes. However, variations in these parameters, including lower antibiotic concentrations normalized to the MIC of each strain (5×or 10×MIC; see *Figure 1—figure supplement 6*), additional washing steps to ensure complete removal of antibiotics (see *Figure 1—figure supplement 6*), and extended agar plate incubation times of up to 48 hours (see *Figure 1—figure supplement 7*), did not affect the persistence phenotype. The Δ*crp* and Δ*cyaA* strains consistently showed reduced ampicillin and ofloxacin persistence compared to WT, whereas all three strains showed no detectable persisters following gentamicin treatment (*Figure 1—figure supplements 6 and 7*). Moreover, we examined the impact of Crp/cAMP disruption in the HipA7 strain to assess whether the role of Crp/cAMP in persistence extends to a genetically sensitized, persister-enriched background. Deletion of either *crp* or *cyaA* in the HipA7 background significantly reduced persistence to both ampicillin and ofloxacin during the late stationary phase (*Figure 1—figure supplement 8*), mirroring the effects observed in the WT strain. Altogether, these results show that disruption of the Crp/cAMP regulatory network impairs persister formation during the late stationary phase, underscoring its critical role in ampicillin and ofloxacin tolerance.

## Crp/cAMP complex governs *E. coli* stationary phase metabolism

To determine whether the reduced ampicillin and ofloxacin persister levels at the late stationary phase in the mutant strains (*Figure 1A and B*) are linked to stationary phase metabolism, we utilized untargeted mass spectrometry (MS). This approach facilitated the quantification of metabolites in Δ*crp* cells, allowing for a comparison with WT controls in both the early and late stationary phases (*Figure 2A*). Since both Δ*crp* and Δ*cyaA* strains exhibit the same persistence phenotype, and *crp* deletion effectively abolishes Crp/cAMP complex function, we used the Δ*crp* mutant for metabolomic profiling to capture key regulatory changes while maintaining experimental feasibility. The metabolomics data were subjected to unsupervised hierarchical clustering, and metabolites identified in independent biological replicates of each strain and condition were found to cluster together (*Figure 2A*), thus confirming the reproducibility of our data.

To elucidate the upregulated and downregulated metabolic pathways in the mutant strain as compared to the WT strain, we performed enrichment analyses utilizing MetaboAnalyst (*Lu et al., 2023*). For downregulated pathways, we considered a threshold ratio of 0.5 or lower, where the ratio indicates metabolite levels in the mutant strain relative to the WT (*Supplementary file 2A*).

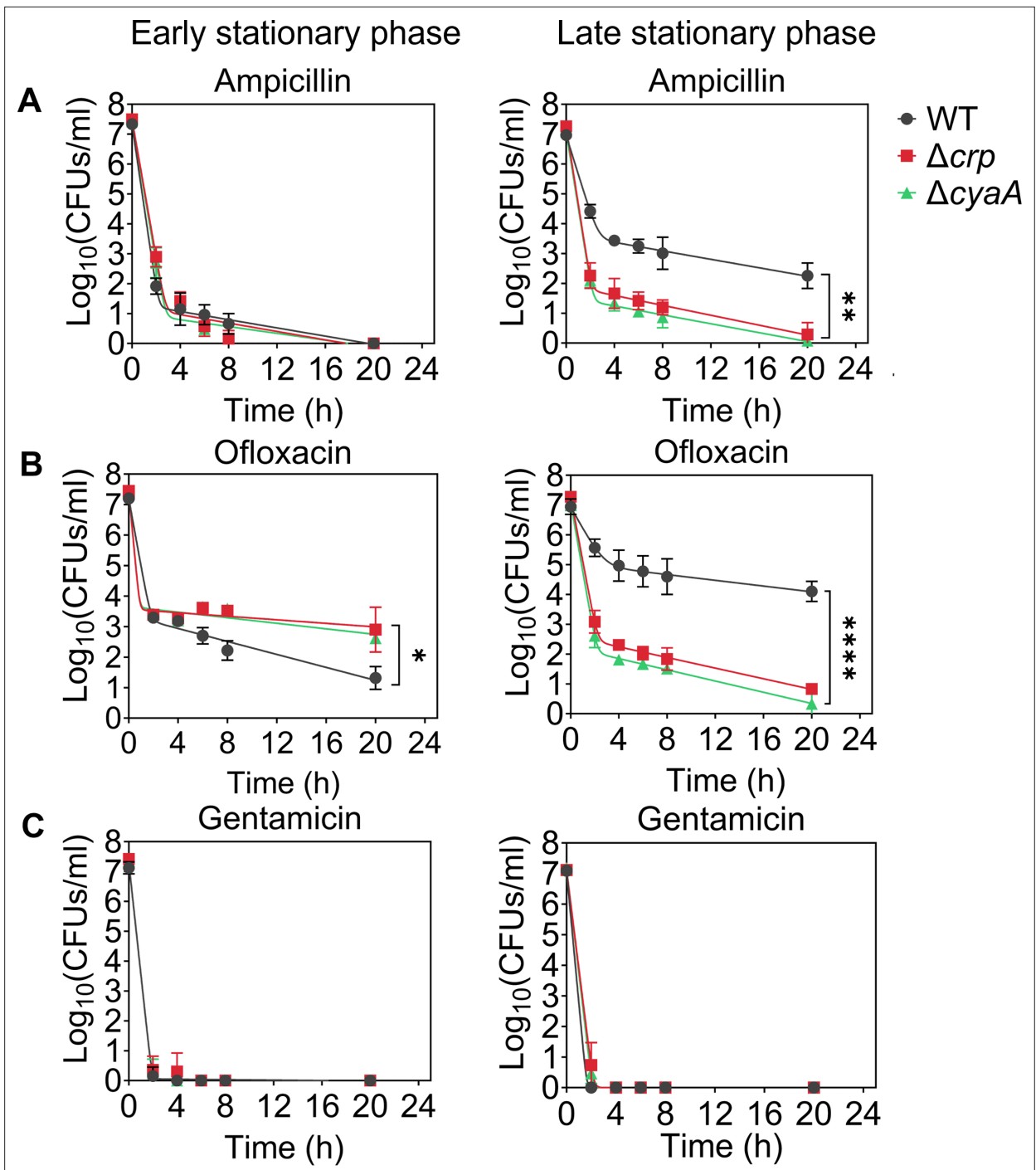

**Figure 1.** Crp/cAMP regulation of persister cell formation in the stationary phase. *E. coli* K-12 MG1655 WT and mutant cells at early (t=5 hr) and late (t=24 hr) stationary phases were transferred to fresh medium with antibiotics for persister cell quantification. At time points 0, 2, 4, 6, 8, and 20 hr, 1 mL of the treated culture was washed with 1 X phosphate-buffered saline (PBS) to remove antibiotics. It was then serially diluted and plated on an agar plate to count the colony-forming units (CFUs). (**A**) Persister levels of ampicillin-treated culture with an antibiotic concentration of 200 µg/mL. (**B**) Persister levels of ofloxacin-treated culture with an antibiotic concentration of 5 µg/mL. (**C**) Persister levels of gentamicin-treated culture with an antibiotic concentration of 50 µg/mL. The number of biological replicates is n=4 for all panels. Biphasic kill curves were generated using a non-linear model (see Materials and methods). Statistical significance tests were conducted using F-statistics (*$p < 0.05$, **$p < 0.01$, ****$p < 0.0001$). The data for each time point represent the mean value ± standard deviation.

The online version of this article includes the following figure supplement(s) for figure 1:

**Figure supplement 1.** cAMP concentrations normalized to cell numbers for *E. coli* K-12 MG1655 WT, Δ*crp*, and Δ*cyaA*.

*Figure 1 continued on next page*

**Figure supplement 2.** Genetic perturbation of Crp/cAMP enhanced $P_{cyaA}$ promoter activity, resulting in increased *gfp* expression.

**Figure supplement 3.** Growth curves of *E. coli* K-12 MG1655 WT, Δ*crp*, and Δ*cyaA*.

**Figure supplement 4.** Normalized cAMP concentrations of *E. coli* K-12 MG1655 WT.

**Figure supplement 7.** Agar plates showing *E. coli* K-12 MG1655 WT, Δ*crp*, and Δ*cyaA* strains following treatment with antibiotics at normalized concentrations.

**Figure supplement 8.** Deletion of *crp* and *cyaA* reduces ampicillin and ofloxacin persistence in the *hipA7* strain.

Conversely, for upregulated pathways, the threshold ratio was set at 2 or higher (*Supplementary file 2B*). The enrichment ratio for each pathway was calculated based on the number of metabolite hits compared to the expected hits derived from the chemical structure library (*Lu et al., 2023*). Our extensive comparison of the mutant cells to the WT cells through pathway enrichment analysis (refer to *Figure 2B*, *Figure 2—figure supplements 1–3* for pairwise comparison of different conditions) revealed several important findings:

1. During the early stationary phase, we observe a slight downregulation in the abundance of TCA cycle metabolites, including citrate and fumarate, in the Δ*crp* strain compared to WT (*Figure 2C*, *Figure 2—figure supplement 1*). However, as the late stationary phase progresses, the downregulation in both TCA cycle and pentose phosphate metabolism becomes more pronounced in the Δ*crp* strain (*Figure 2B and D*).
2. The Δ*crp* strain exhibits upregulation of several metabolites compared to WT during both early and late stationary phases, primarily associated with anabolic pathways. Particularly, the upregulation of some of these pathways becomes more pronounced during the late stationary phase. These pathways include crucial metabolites like deoxyribonucleosides and ribonucleosides (which play essential roles in DNA and RNA synthesis), fatty acids and carboxylic acids (the main components of bacterial cell membranes), and peptides (which are linked to protein synthesis) (*Figure 2B*).
3. During the early stationary phase, we noticed a significant upregulation in the abundance of intermediate metabolites related to glycolysis, gluconeogenesis, and pyruvate metabolism in mutant cells compared to WT cells (*Figure 2C*). This observation is not surprising, as the inhibition of the TCA cycle in the mutant strain could potentially redirect metabolic fluxes toward glycolysis and lactate metabolism.

Altogether, our metabolic data indicate that, in the stationary phase, WT cells maintain their energy metabolism to some extent while downregulating their anabolic pathways (*Figure 2A*). This metabolic state appears to be regulated by the Crp/cAMP complex, as perturbing its function leads to a significant downregulation of energy metabolism and an upregulation in the abundance of anabolic metabolites (*Figure 2B*).

## Proteomics analysis revealed upregulated pathways in the Δ*crp* strain associated with anabolic metabolism, alongside the downregulation of key proteins in energy metabolism

Since the Crp/cAMP complex acts as a transcriptional regulator affecting the expression of metabolic proteins whose abundance directly affects cellular metabolites, we performed untargeted proteomics, our second genomic-level study, to further validate our results. Considering the noticeable metabolic alterations observed during the late stationary phase, we utilized MS to quantify proteins in Δ*crp* cells and compared them to WT controls at this stage. The resulting proteomics data were subjected to unsupervised hierarchical clustering, and the proteins identified in independent biological replicates of each strain were found to cluster together, confirming the consistency of our findings (*Figure 3—figure supplement 1*). By analyzing protein-protein association networks and employing functional enrichment through STRING (*Szklarczyk et al., 2021*; *Szklarczyk et al., 2023*), which integrates various functional pathway classification frameworks such as Gene Ontology annotations, KEGG pathways, and UniProt keywords, we pinpointed various upregulated and downregulated pathways in the mutant strain when compared to the WT (*Supplementary file 3*). The upregulated pathways are

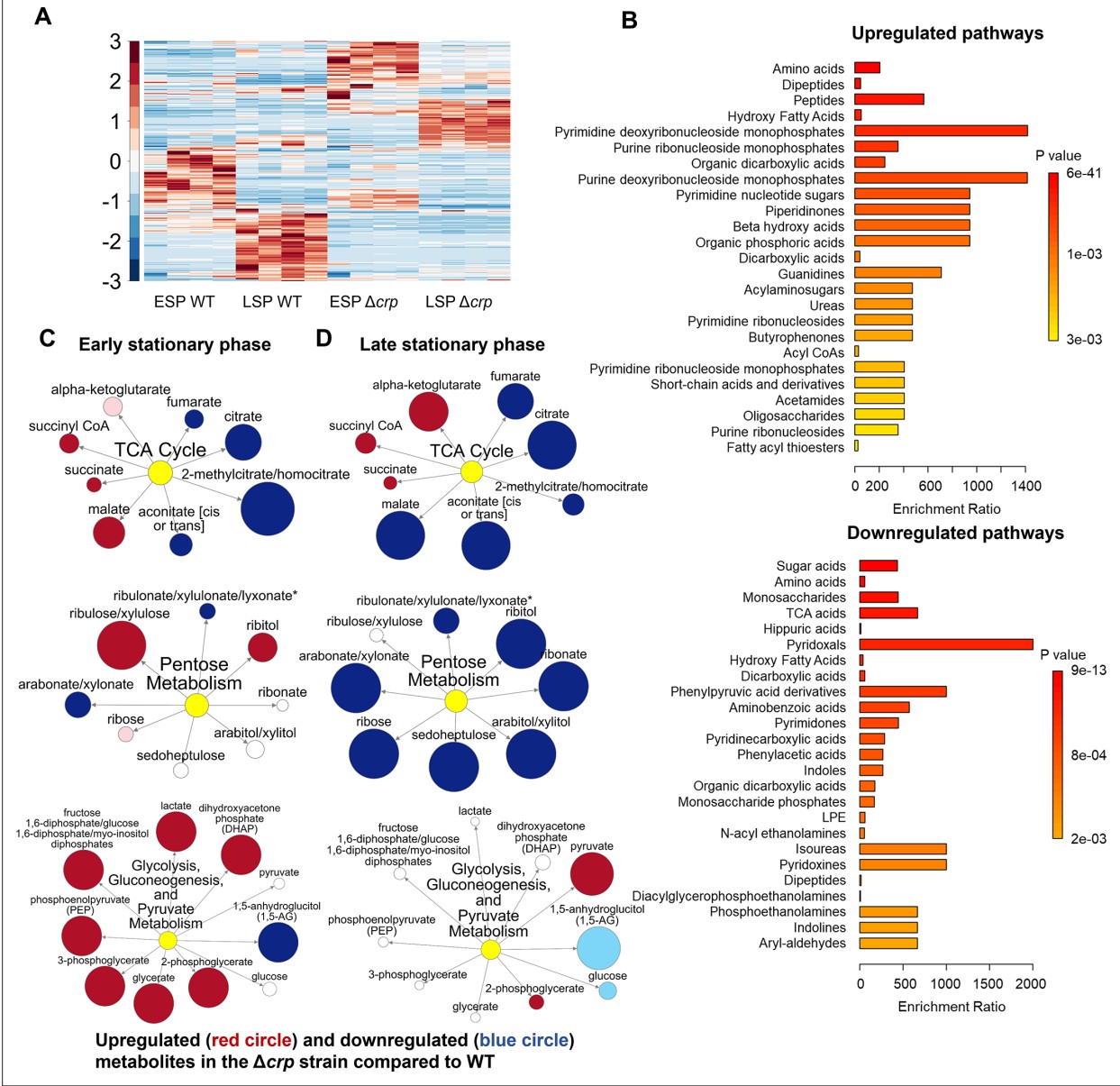

**Figure 2.** The effect of Crp/cAMP on persister cell metabolism during stationary phase. (**A**) MS analysis of *E. coli* K-12 MG1655 WT, and Δ*crp* at early (t=5 hr) and late (t=24 hr) stationary phases. Unsupervised hierarchical clustering was applied to standardized metabolic data. Each column represents a biological replicate. n=4. (**B**) Pathway enrichment analysis was conducted using MetaboAnalyst (*Lu et al., 2023*). Upregulated and downregulated pathways of the Δ*crp* strain compared to WT in the late stationary growth phase were provided in this figure. (**C, D**) Pathway enrichment maps comparing metabolites of the TCA cycle, pentose phosphate metabolism, glycolysis, gluconeogenesis, and pyruvate metabolism in Δ*crp* versus WT for early and late stationary phase conditions, respectively. Circle size corresponds to the ratio of normalized metabolite intensities between mutant and control cells. Blue (p≤0.05 for dark blue; 0.05<p < 0.10 for light blue) and red (p≤0.05 for dark red; 0.05<p < 0.10 for light red) indicate significantly downregulated or upregulated metabolites in the mutant compared to the control. White signifies no significant difference. n=4 for all panels. ESP: Early stationary phase, LSP: Late stationary phase.

The online version of this article includes the following source data and figure supplement(s) for figure 2:

**Source data 1.** Normalized metabolomics data of early and late stationary phases of wild-type and mutant *crp* across three biological replicates.

**Figure supplement 1.** The pathway enrichment analysis comparing WT and Δ*crp* strains during the early stationary phase cultures.

**Figure supplement 2.** The pathway enrichment analysis for the WT strain.

**Figure supplement 3.** The pathway enrichment analysis for the Δ*crp* strain.

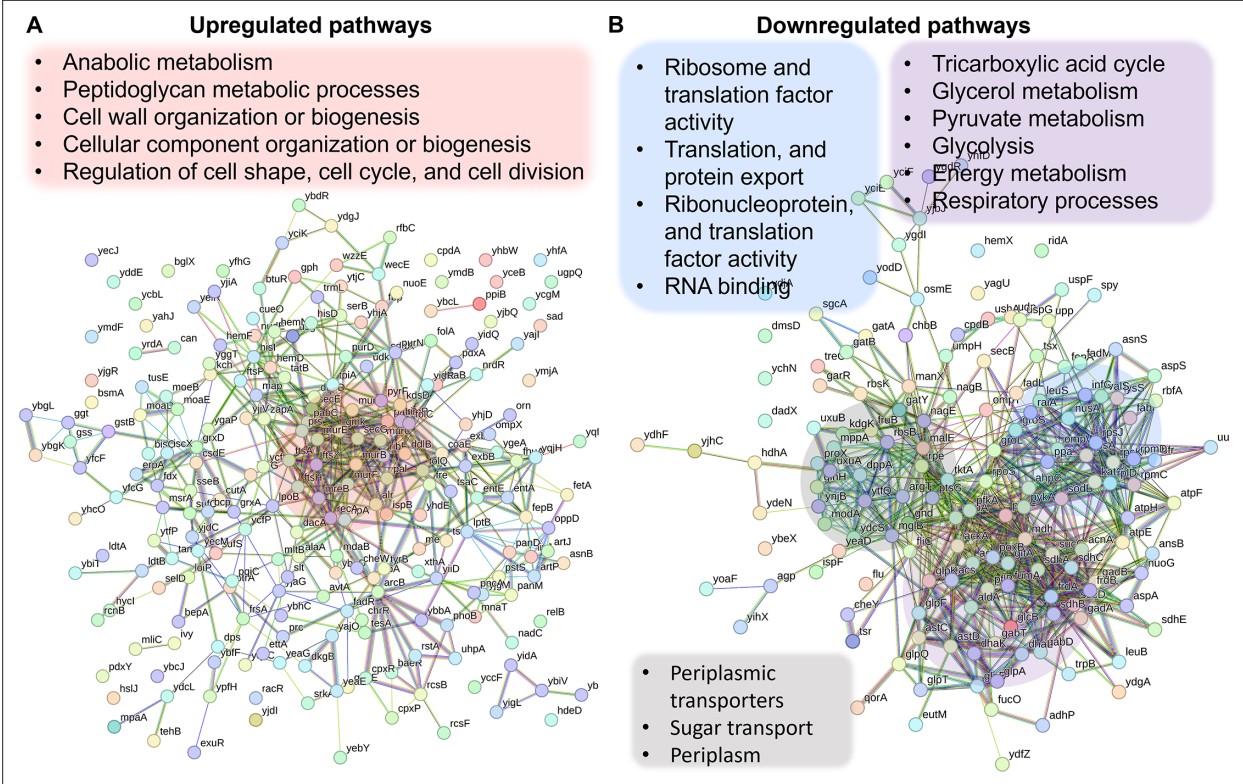

**Figure 3.** Validation of Crp/cAMP-mediated metabolic state in persister cells through proteomics analysis. Pathway enrichment analysis was conducted in STRING (*Szklarczyk et al., 2021*; *Szklarczyk et al., 2023*) for upregulated (**A**) and downregulated (**B**) proteins. Genes highlighted in red are linked with the upregulated protein networks, while genes in blue, gray, and purple correspond to those in the downregulated protein network. The visual network in STRING illustrates protein interactions. In evidence mode, color in the network represents the interaction evidence of data support, derived from curated databases, experimental data, gene neighborhood, gene fusions, co-occurrence, co-expression, protein homology, and text mining (*Szklarczyk et al., 2021*; *Szklarczyk et al., 2023*).

The online version of this article includes the following source data and figure supplement(s) for figure 3:

**Source data 1.** Normalized proteomics data of the late stationary phase of wild-type and mutant *crp* across three biological replicates.

**Figure supplement 1.** The MS analysis of proteins from both WT and Δ*crp* strains at the late stationary phase.

associated with anabolic metabolism and encompass peptidoglycan metabolic processes, cell wall organization or biogenesis, cellular component organization or biogenesis, regulation of cell shape, cell cycle, and cell division (*Figure 3A*). Also, the mutant strain displayed downregulated pathways, encompassing glycerol metabolism, TCA cycle, pyruvate metabolism, glycolysis, and various pathways associated with ribosome and transcriptional factor activity (*Figure 3B*). Our analysis specifically pinpointed a cluster of proteins involved in energy metabolism and respiratory processes. Notably, this cluster includes GltA, SdhB, SucC, SucD, FrdB, FrdA, AcnA, AceA, and Mdh proteins, which play crucial roles in either the TCA cycle or as membrane-bound components of ETC (*Figure 3B*). Altogether, the alignment between metabolomics and proteomics analyses provides additional validation for the Crp/cAMP-mediated metabolic state.

## Crp/cAMP complex shapes *E. coli* cell proliferation dynamics

The omics data suggest that the upregulation in the abundance of anabolic metabolites and proteins, particularly those related to cell wall organization or biogenesis, cell cycle, and cell division, in the stationary-phase mutant cells, would likely enhance their ability to resume growth upon transitioning to fresh medium. To investigate this, we utilized a cell proliferation assay that employed an inducible fluorescent protein (mCherry) expression cassette. This assay provided us with the ability to monitor non-growing cells at a single-cell resolution, as described previously (*Orman and Brynildsen, 2013a*; *Roostalu et al., 2008*). The mCherry expression cassette is controlled by an isopropyl

ß-D-1-thiogalactopyranoside (IPTG) inducible synthetic T5 promoter that was previously inserted into the chromosome of an *E. coli* strain carrying a *lacI^q* promoter mutation (***Orman and Brynildsen, 2013a***). This configuration allowed for precise regulation of mCherry expression using IPTG. Here, we introduced *crp* and *cyaA* deletions into this strain. These deletions reduced the persistence of the mCherry-expressing *E. coli* strain in the late stationary phase (***Figure 4—figure supplement 1***), consistent with the findings presented in ***Figure 1***. To perform the growth assay, we induced mCherry expression in the main cultures and then washed the cells to remove IPTG. The cells were then inoculated into a fresh medium without IPTG, and their growth was analyzed with a flow cytometer. This allowed us to track the dilution of mCherry protein within the cells, which served as an indicator of cell proliferation. As shown in ***Figure 4A***, initially, all cells exhibited high red fluorescence. However, as cells underwent division, the red fluorescence of the overall population decreased in the absence of the inducer (***Figure 4A***). Notably, within the WT strain, a subpopulation from the late stationary phase cultures displayed constant fluorescence levels, indicating their inability to divide (***Figure 4B***). In contrast to the WT strain, we did not detect similar subpopulations in the mutant strains (***Figure 4B***). Additionally, this subpopulation of non-growing cells does not emerge during the early stationary phase cultures (***Figure 4A***). This observation provides an explanation for the observed reduction in persister levels in these mutant strains in the late stationary phase, as the enrichment of persister cells within these non-growing cell subpopulations was reported in previous studies (***Jõers et al., 2010***; ***Orman and Brynildsen, 2013b***; ***Roostalu et al., 2008***).

To confirm the significance of Crp/cAMP in the formation of non-growing cells, we introduced the expression plasmid carrying the *crp* gene into the Δ*crp* strain. As anticipated, the introduction of the *crp* expression plasmid resulted in the emergence of a non-growing population within the culture, contrasting with the mutant strain containing the empty plasmid vector used as a control (***Figure 4—figure supplement 2A, B***). The reduced capacity of stationary-phase WT cells to initiate proliferation upon transfer to a fresh medium suggests the possible presence of an extended lag phase in these cells. To investigate this, we employed flow cytometry to precisely quantify cell numbers and generate growth curves for both WT and mutant strains. As anticipated, the growth curve of the WT strain displayed a slower initial growth rate and a prolonged lag phase duration compared to the mutant strains (***Figure 4C***). Conversely, the mutant strains displayed a shorter lag phase, yet they demonstrated an increased doubling time in the exponential phase compared to WT (***Figure 4C***), which was also anticipated, considering their decreased reliance on oxidative phosphorylation due to TCA cycle inhibition. Altogether, these results provide additional support and validation for the findings from our metabolomics and proteomics data, as our data reveals a correlation between the abundance of molecules associated with cell division and the ability of the stationary phase cells to resume growth.

## Persister cells rely on energy metabolism

The Crp/cAMP-mediated metabolic state, characterized by increased respiration in WT compared to mutant strains, was further validated using redox sensor green (RSG) dye and a reporter plasmid measuring the promoter activity of succinate:quinone oxidoreductase (SQR) genes. The SQR reporter system (***Zaslaver et al., 2006***) employs green fluorescent protein (GFP) expression, regulated by the promoter of the SQR operon, which includes the *sdhA*, *sdhB*, *sdhC*, and *sdhD* subunit genes. The SQR complex plays a vital role in cellular metabolism by catalyzing the oxidation of succinate to fumarate concurrently with the reduction of ubiquinone to ubiquinol, thus directly linking the TCA cycle with the respiratory ETC (***Keseler et al., 2005***). Our results indicate an upregulation of the SQR promoter activity in WT cells compared to the mutant strains in the stationary phase, validating the findings from our metabolomics and proteomics data (***Figure 5A***). The RSG dye, on the other hand, serves as a well-established metabolic indicator, measuring bacterial oxidation-reduction activity, a crucial function involving the ETC driven by the TCA cycle. Once reduced by bacterial reductases, the RSG dye emits a stable green fluorescent signal (***Figure 5—figure supplement 1***). Our data demonstrate that the redox activities of WT cells are much higher and more heterogeneous compared to those of the mutant strains in the late stationary phase, further corroborating the results from our preceding analyses (***Figure 5B***). Furthermore, we utilized a methodology that integrates the mCherry expression system, flow cytometry, and ampicillin-mediated cell lysis to determine whether persister cells in WT still maintain their respiration. In this assay, both the WT and mutant strains carrying the mCherry expression system were exposed to ampicillin after transferring them to a fresh medium. The

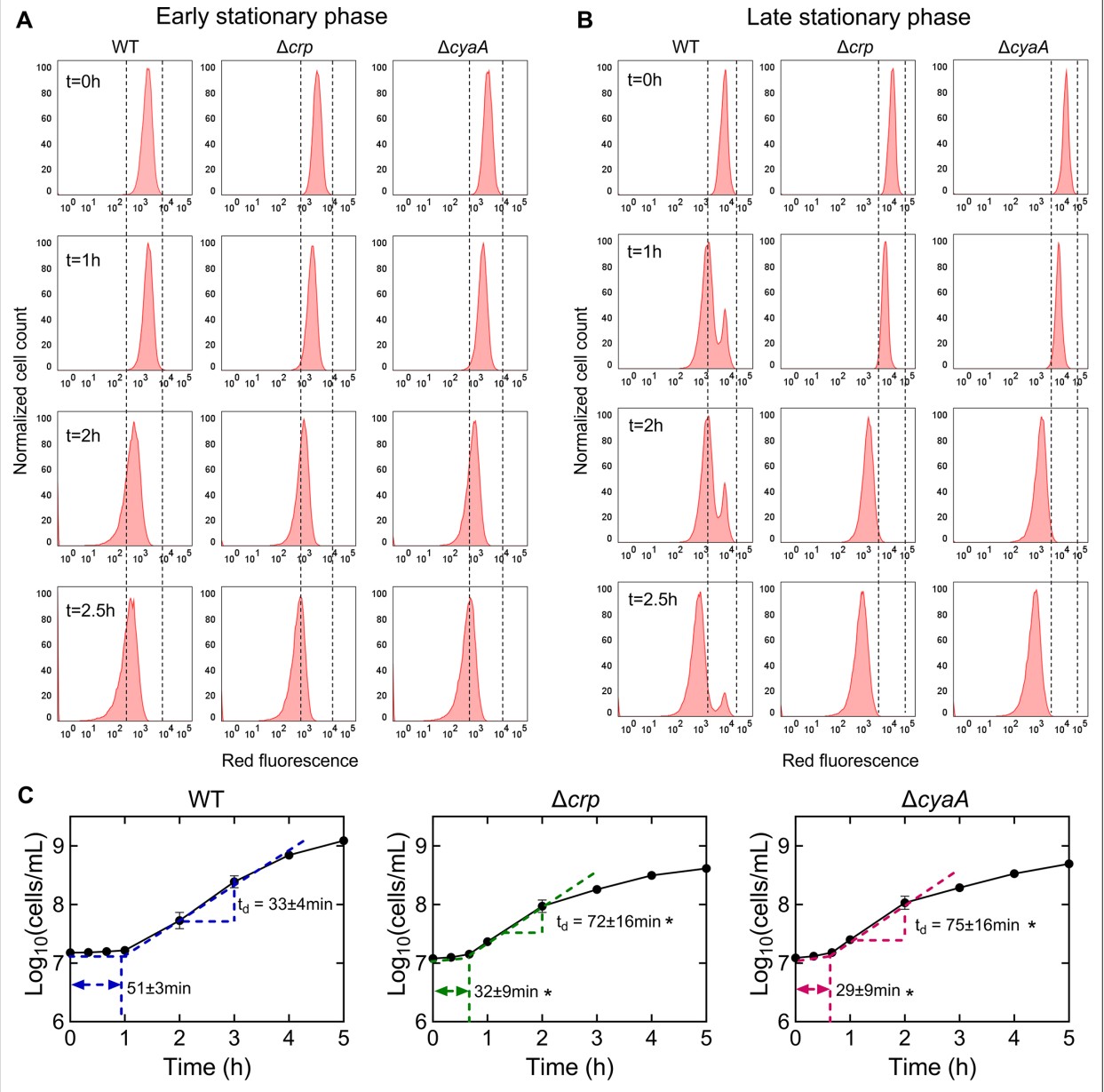

**Figure 4.** The role of Crp/cAMP in non-growing cell formation. (**A, B**) Flow cytometry histograms depict mCherry expression in *E. coli* K-12 MG1655 WT, Δ*crp*, and Δ*cyaA* at early (t=5 hr) and late (t=24 hr) stationary phases, respectively. Cells containing an IPTG-inducible mCherry expression system were cultivated with IPTG. After washing and dilution of early and late stationary phase cells in IPTG-free fresh media, fluorescence was tracked in non-growing and growing cells for 2.5 hr. The panel is a representative biological replicate. Consistent results were seen across all three biological replicates. (**C**) Growth curves of WT, Δ*crp*, and Δ*cyaA* cultures were determined using flow cytometry to calculate lag and doubling times. Lag times were calculated using the 'Microbial lag phase duration calculator' (**Opalek et al., 2022**). Doubling times were computed using the formula $t_d=\Delta t/(3.3 \times Log_{10}(N/N_o))$. n=3. *Statistical significance observed between control and mutant strains (p<0.05, two-tailed t-test). The data for each time point represent the mean value ±standard deviation.

The online version of this article includes the following figure supplement(s) for figure 4:

**Figure supplement 1.** Persister levels of *E. coli* WT, Δ*crp*, and Δ*cyaA* cells with the integrated mCherry expression system.

**Figure supplement 2.** Non-growing cell levels in the *E. coli* strain carrying the Crp expression system.

inducer was added to both growth and treatment cultures to sustain the cells' red signals. Unlike other antibiotics, ampicillin disrupts cell wall synthesis, leading to the lysis of cells upon their resumption of growth. As seen in *Figure 5—figure supplement 2*, the cells that were lysed lost their mCherry signals. On the other hand, the resilient, tolerant cells that evaded ampicillin-induced lysis maintained

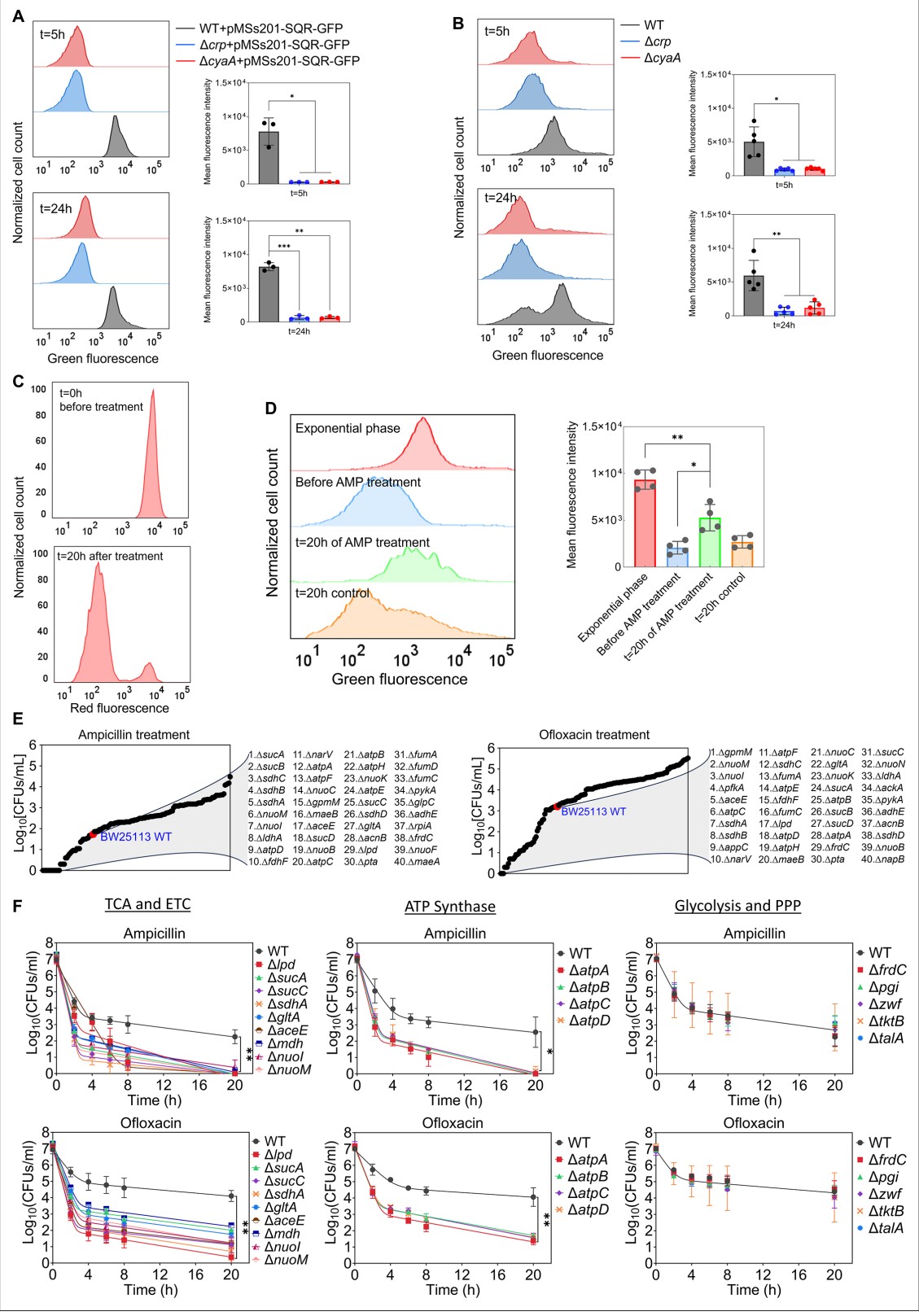

**Figure 5.** Crp/cAMP-mediated metabolic state of persister cells. (**A**) GFP reporter plasmid introduced into *E. coli* K-12 MG1655 WT, Δ*crp*, and Δ*cyaA* cells to monitor SQR gene activity. Flow cytometry was used to detect activity at early (t=5 hr) and late (t=24 hr) stationary phases. The panel on the left represents a biological replicate, and the results are consistent across all three replicates, as demonstrated in the panel on the right. Statistical significance observed between control and mutant groups (*p<0.05, **p<0.01, ***p<0.001, two-tailed t-test). (**B**) Redox activities of *E. coli* K-12 MG1655

*Figure 5 continued on next page*

*Figure 5 continued*

WT, Δ*crp*, and Δ*cyaA* cells were measured at early (t=5 hr) and late (t=24 hr) stationary phases by flow cytometry using a RSG dye. This dye fluoresces green after reduction by bacterial reductases. A representative biological replicate is shown (left), with consistent results across all five replicates (right). Statistical significance observed between control and mutant groups (*p<0.05, **p<0.01, two-tailed t-test). (**C**) *E. coli* cells with integrated mCherry expression system used to validate cellular respiration. Cells were diluted into fresh media and treated with ampicillin (200 μg/mL) for 20 hr. Flow cytometry measured the red fluorescence of intact surviving cells. A representative biological replicate is shown, with consistent results across all three replicates. (**D**) RSG levels of cells (carrying the mCherry expression system) at exponential phase (t=3 hr); cells before ampicillin treatment; non-lysed (intact) cells after 20 hr of ampicillin treatment; and untreated cells after 20 hr of culturing. A representative biological replicate is shown (left), with consistent results across all four replicates (right). Statistical significance observed between intact antibiotic-treated cells and others (*p<0.05, **p<0.01, two-tailed t-test). (**E**) High-throughput screening of mutants from the Keio collection. The mutant strains selected are associated with central metabolism. Stationary phase cells were diluted 100-fold in fresh medium and treated with ampicillin (200 μg/mL) or ofloxacin (5 μg/mL) for 20 hr. Treated cultures were washed, serially diluted, and plated on agar plates to quantify CFUs. (**F**) Genes related to the TCA cycle, ETC, ATP synthase, glycolysis, and pentose phosphate pathway (PPP) were knocked out and then treated with ampicillin (200 μg/mL) or ofloxacin (5 μg/mL) to enumerate CFUs. n=4. Biphasic kill curves were generated using a non-linear model. Statistical significance tests were conducted using F-statistics (*p<0.05, and **p<0.01). Each data point represents the mean value ± standard deviation.

The online version of this article includes the following figure supplement(s) for figure 5:

**Figure supplement 1.** RSG staining control for bacterial metabolic activities.

**Figure supplement 2.** Intact (non-lysed) cell levels of *E. coli* WT, Δ*crp*, and Δ*cyaA* cells with the integrated mCherry expression system.

**Figure supplement 3.** VBNC levels of *E. coli* WT, Δ*crp*, and Δ*cyaA* cells with the integrated mCherry expression system.

**Figure supplement 4.** Cell counts of *E. coli* K-12 MG1655 WT and mutant strains using flow cytometry at late stationary phase.

their mCherry levels throughout the treatment. In the mutant strains, ampicillin was effective in lysing almost all cells as anticipated, leaving no or a small number of intact cells (***Figure 5—figure supplement 2***). However, in the WT strain, we detected a subpopulation of intact cells throughout the entire treatment period (***Figure 5C***, ***Figure 5—figure supplement 2***). While the population-level redox activities of these tolerant intact cells in WT are lower than those of exponential phase cells, they still displayed a significant increase in RSG levels compared to cell populations before antibiotic treatments or untreated control cells subjected to identical conditions (***Figure 5D***)**,** suggesting that they maintain steady-state energy metabolism. We want to highlight that not all intact cells in the WT strain reported here are persisters. A significant portion comprises 'viable but non-culturable' (VBNC) cells, and WT cells exhibit markedly higher VBNC levels than Δ*crp* and Δ*cyaA* strains (***Figure 5—figure supplement 3***). VBNC cells can be quantified from intact cells following beta-lactam treatments (***Orman and Brynildsen, 2013b***; ***Roostalu et al., 2008***). These cells may exhibit metabolic activities but are unable to readily colonize upon transfer to fresh medium (***Ayrapetyan et al., 2018***; ***Orman and Brynildsen, 2013b***). Collectively, our metabolic measurement data (***Figure 5A–C***) aligns with the findings from our omics analyses, and the number of intact cells observed in WT after beta-lactam treatment is consistent with the count of non-growing cells in WT (***Figure 4B*** vs ***Figure 5—figure supplement 2***). These non-growing cells are anticipated to be less susceptible to lysis by beta-lactams (***Orman and Brynildsen, 2013b***; ***Roostalu et al., 2008***).

## The genomic-level screening of *E. coli* knockout underscores the significance of energy metabolism in sustaining the viability of persister cells

Although some metabolic genes, including those encoding the TCA cycle (e.g. *sdhA*, *sucB*, *mdh*, *icd*), have been studied in *E. coli* (***Luidalepp et al., 2011***; ***Ma et al., 2010***; ***Manuse et al., 2021***; ***Orman and Brynildsen, 2015***; ***Yu et al., 2019***), a comprehensive genomic-level screening strategy is necessary to validate which metabolic pathways are truly associated with antibiotic tolerance. To further underscore the importance of energy metabolism, we conducted a high-throughput screening of 149 different *E. coli* K-12 BW25113 mutant strains from the Keio knockout library (***Baba et al., 2006***). The selected strains are related to central carbon metabolism, encompassing glycolysis, pentose phosphate pathways, TCA cycle, ETC, ATP synthase, and fermentation pathways (***Figure 5E***). While the deletion of genes related to glycolysis and pentose phosphate pathways did not affect antibiotic tolerance in the cells, some mutant strains associated with cytochrome bo and quinone oxidoreductase complexes (e.g. *cyo* genes and *nuoL*) exhibited enhanced tolerance (***Supplementary file 4***). However, the mutant strains exhibited reduced tolerance to both antibiotics compared to the control

*E. coli* K-12 BW25113 WT strain, which was found to be largely associated with the TCA pathway (*sucA, sucB, lpd, sucC, sucD, sdhA, sdhB, sdhC, sdhD, gltA, acnB, aceE, fumA, mdh* and *fumC*), ETC (*nuoB, nuoC, nuoI, nuoK, nuoM,* and *narV*), ATP synthesis (*atpA, atpB, atpC, atpD, atpE,* and *atpH*), and mixed acid fermentation pathways (*ldhA, fdhF, pta, adhE,* and *frdC*) (*Figure 5E*, *Supplementary file 4*). We acknowledge that the Keio strains were generated in a high-throughput manner, and there might be unknown errors in their genomic DNA. To ensure the reproducibility of our findings, we generated knockout strains for key genes associated with the TCA, ETC, ATP synthase, glycolysis, and pentose phosphate pathway (*Figure 5F*, see *Supplementary file 5* for detailed description of genes). We then tested their antibiotic tolerance, and our results were consistent with the omics data and screening outcomes (*Figure 5F*), confirming the critical role of energy metabolism, specifically the TCA cycle, ETC, and ATP synthase, in bacterial persistence. We note that equal numbers of cells from each strain were transferred into antibiotic treatment media to ensure consistency in cell numbers, and no significant growth deficiencies or differences in cell density were observed in the late stationary phase of the knockout strains (*Figure 5—figure supplement 4*).

## Discussion

Our study highlights the crucial role of the Crp/cAMP complex in maintaining the metabolic state of stationary-phase persister cells, enabling their survival under adverse conditions. Through metabolomics, proteomics, and high-throughput screening of single-gene deletion strains, we substantiated that the Crp/cAMP regulatory complex sustains an active respiratory state while downregulating anabolic pathways in persister cells. This respiratory state is vital for the survival of persister cells, as perturbing the Crp/cAMP complex or respiration significantly reduced persister phenotypes, which may explain the previously reported decrease in antibiotic tolerance in *E. coli* cells cultured under anaerobic conditions (*Orman and Brynildsen, 2015*). Notably, we observed an upregulation of anabolic metabolites and proteins when the Crp/cAMP regulatory complex was perturbed, particularly those associated with cell wall organization, cell cycle, and cell division, enhancing the ability of stationary-phase mutant cells to resume growth. Although the literature has shown associations between antibiotic tolerance with proteins involved in cell division and the TCA cycle (e.g. SdhA, SucB, Mdh) (*Luidalepp et al., 2011*; *Ma et al., 2010*; *Orman and Brynildsen, 2015*; *Yu et al., 2019*), our study establishes a strong link between these critical cellular processes and the Crp/cAMP complex, providing much-needed clarity in the field. In fact, upon investigating Crp/cAMP regulons via the Ecocyc database (*Keseler et al., 2005*), we identified that certain metabolic genes, deleted in our *E. coli* K-12 MG1655 background, are potentially regulated by Crp/cAMP (*Supplementary file 5*), providing additional support for the validity of our omics results.

We acknowledge that our metabolomic and proteomic data were obtained at the whole-population level, rather than from isolated persister cells. Fluorescent reporters combined with fluorescence-activated cell sorting (FACS) have been utilized to study persister cells, including in our previous studies (*Amato et al., 2013*; *Orman and Brynildsen, 2013a*; *Orman and Brynildsen, 2015*). However, this approach only enriches for persisters rather than isolating a pure population, as persisters still constitute a small fraction of the sorted cells. Despite these limitations, our population-level analyses provide valuable insights into the role of the Crp/cAMP complex in regulating non-growing cell formation and persistence (*Figure 4*). We further validated these findings through single-gene deletions and flow cytometry-based assays (*Figure 5*), confirming a consistent reduction in both non-growing and persister cells in Crp/cAMP-deficient strains.

The deletion of *cyaA* resulted in a reduction of cAMP levels, as expected given the role of the CyaA enzyme in cAMP synthesis (*Figure 1—figure supplement 1*). Conversely, the removal of *crp* led to an increase in cAMP levels compared to those in wild-type cells. Notably, this increase is statistically significant (*Figure 1—figure supplement 1*). Although this is an interesting observation, it is likely due to the feedback regulation of the Crp/cAMP complex on *cyaA* expression (*Aiba, 1985*; *Keseler et al., 2011*; *Majerfeld et al., 1981*). Specifically, disruption of Crp function is expected to derepress the *cyaA* promoter (P$_{cyaA}$), leading to increased CyaA expression and elevated cAMP levels. To test this, we employed a P$_{cyaA}$-*gfp* reporter plasmid (pMSs201) in *E. coli* K-12 MG1655 WT, Δ*crp*, and Δ*cyaA* strains to monitor promoter activity. As anticipated, deletion of *crp* led to enhanced *gfp* expression, confirming that Crp/cAMP negatively regulates *cyaA* expression via feedback. These results clarify the

observed cAMP differences and support the regulatory role of the Crp/cAMP complex (*Figure 1—figure supplement 2*).

Our findings reveal substantial heterogeneity in metabolism (measured by RSG) among WT stationary phase cells, contrasting with the more uniform behavior observed in Δ*crp* and Δ*cyaA* strains (*Figure 5B*). We also demonstrated the presence of two distinct populations in WT cells during the late stationary phase: one that resumes rapid growth and another subpopulation that does not resume growth when transferred to a fresh medium (*Figure 4B*). The absence of this non-growing cell subpopulation in the mutant strains could account for their sensitivity to AGs. However, the mechanism by which AGs kill these non-growing cells in the WT strain remains perplexing. The underlying reasons might be linked to their metabolism, as AG uptake is an energy-requiring process, relying on the electron flow through membrane-bound respiratory chains (*Allison et al., 2011*). Moreover, persister cells obtained from various antibiotics, such as ampicillin and ofloxacin, in WT *E. coli* were previously found to exhibit sensitivity to AGs when sugar molecules were introduced into the cultures (*Allison et al., 2011*; *Orman and Brynildsen, 2013b*). However, the enhanced sensitivity mediated by sugar molecules was reversed to its original state in a subsequent study when the Crp/cAMP complex was genetically perturbed (*Mok et al., 2015a*). This can be attributed to the lack of active energy metabolism in these genetically altered strains, as suggested by our comprehensive genomic-level analyses here. While the absence of cell division in non-growing cell subpopulations in WT may suggest a down-regulation in anabolic metabolism, their energy metabolism may remain partially active, which could potentially explain the phenomenon of AG potentiation. Indeed, our results presented in *Figure 5D* support this interpretation.

Antibiotics are generally effective against proliferating bacteria, leading to the notions that tolerance is linked to temporary growth suppression and that persister cells are dormant phenotypes with repressed metabolism. Although persister cells are generally considered to have reduced metabolic activity compared to exponentially growing cells, their survival may still depend on energy metabolism. Also, the direct comparison of persister cell metabolism to that of exponentially growing cells may not be the best approach. Growing cells have a very high energy output and consume metabolites at a fast pace. Therefore, any comparison between tolerant and non-tolerant cell populations requires proper normalization techniques such as adjusting cellular metabolic activities to the amount of substrate utilized by cells. An example of this normalization was conducted by Heinemann's group (*Radzikowski et al., 2016*), demonstrating that ATP production rates per substrate in tolerant cells exceed those of exponentially growing cells.

We diluted cells in fresh medium at early and late stationary phases before antibiotic treatments. This step is essential for quantifying type I persisters, as these cells do not readily resume growth upon dilution in the fresh medium during the lag phase (*Balaban et al., 2004*; *Orman and Brynildsen, 2015*). We acknowledge that antibiotic tolerance is influenced by various factors, including culture dilutions, media, specific strains, antibiotics tested, treatment durations, and the growth phase during treatment administration. These factors may contribute to variations in reported persister levels observed in the Δ*cyaA* strain during the exponential phase (*Chu et al., 2012*; *Molina-Quiroz et al., 2018*; *Parsons et al., 2024*; *Sulaiman and Lam, 2020*; *Yamasaki et al., 2020*; *Zeng et al., 2022a*). To assess how *cyaA* and *crp* deletions affect antibiotic responses under conditions similar to those used by Zeng et al. (*Zeng et al., 2022b*) —specifically, exponential-phase *E. coli* BW25113 strains (Keio collection), lower antibiotic concentrations, and short treatments (e.g. 1 hr)—we first tested *E. coli* MG1655 WT, Δ*crp*, and Δ*cyaA* strains in late stationary phase using reduced antibiotic concentrations and shorter exposures. Both knockouts showed decreased survival following ampicillin and ofloxacin treatment compared to WT (*Supplementary file 6*), consistent with our findings in *Figure 1*. In the exponential phase, the knockout strains exhibited reduced survival after ampicillin treatment but increased survival after ofloxacin treatment relative to WT (*Supplementary file 7A*), again mirroring the trends in *Figure 1*. Gentamicin treatment, however, produced variable results in MG1655 knockouts, likely due to the brief 1 hr exposure being insufficient for robust conclusions (*Supplementary file 7A*). Notably, when we tested the corresponding Keio knockout strains in the BW25113 background, we observed increased tolerance in exponential-phase cells, reproducing Zeng et al.'s findings under their specific conditions (*Supplementary file 7B*), although BW25113 and MG1655 exhibited distinct persister phenotypes in exponential phase (*Supplementary file 7A and B*). These results, altogether, highlight the sensitivity of antibiotic

tolerance and persistence phenotypes to factors such as strain background, antibiotic concentration, and treatment duration.

While we did not focus on the exponential growth phase in our study, it is noteworthy that the reduced growth rate during this phase in the mutant strains (*Figure 4C*) may explain the antibiotic tolerance observed in previous studies involving the *cyaA* deletion (*Chu et al., 2012*; *Molina-Quiroz et al., 2018*; *Sulaiman and Lam, 2020*; *Yamasaki et al., 2020*). The growth disparity noted between mutant and WT strains, particularly evident around 5 hr in our results (*Figure 1—figure supplement 3*), may also be linked to the ofloxacin persisters observed in the mutant strains at this specific time point (*Figure 1*). While slow cell growth may indeed correlate with bacterial persistence (*Kaldalu and Tenson, 2019*), it is important to note that the persistence associated with perturbations of metabolic genes cannot be solely attributed to the slow growth. In fact, the persistence of these mutant strains should depend on many factors (see *Supplementary file 5*) as reported by diverse research groups (*Kim et al., 2016*; *Luidalepp et al., 2011*; *Ma et al., 2010*; *Orman and Brynildsen, 2015*; *Pandey et al., 2021*; *Shan et al., 2015*; *Spoering et al., 2006*; *Wang et al., 2018*; *Yu et al., 2019*; *Zalis et al., 2019*). For instance, in *E. coli*, TCA inactivation was shown to decrease ampicillin and ofloxacin persistence during the lag phase (*Orman and Brynildsen, 2015*), yet it enhances gentamicin tolerance in the exponential phase, which remains unexplained by factors such as cell growth, redox activities, proton motive force (PMF), or ATP levels (*Shiraliyev and Orman, 2023*). Furthermore, gene deletions often trigger pleiotropic effects, leading to unique tolerance mechanisms not evident in wild-type strains. In our Keio screening data analysis, we observed that the deletion of *icd* appeared to enhance persistence (*Supplementary file 4*), in line with a previous study (*Manuse et al., 2021*). The *icd* gene encodes a TCA cycle enzyme, isocitrate dehydrogenase. While it remains unclear whether this observed outcome is attributable to other unseen pleiotropic effects stemming from the *icd* deletion, our data consistently indicates that the most significant reduction in persistence levels occurs with disruptions in energy metabolism. A comprehensive approach, encompassing omics and knockout screening as presented in this study, offers a more complete understanding, revealing the consensus behavior within the entire metabolic network.

Reactive oxygen species (ROS) have been proposed to contribute to antibiotic killing; however, our prior work using identical experimental conditions demonstrated that ROS are unlikely to be a major factor in persister formation during the late stationary phase (*Orman and Brynildsen, 2015*). In that study, we overexpressed catalytically active antioxidant enzymes, including catalases (KatE, KatG) and superoxide dismutases (SodA, SodB, SodC), yet observed no significant change in persister levels. To further decouple ROS from respiratory activity in that study, we performed anaerobic experiments using nitrate as an alternative terminal electron acceptor. Interestingly, anaerobic respiration enhanced persister formation, and inhibition of nitrate reductases with KCN reduced it—further supporting an ROS-independent mechanism. Together, these findings suggest that under our experimental conditions, it is respiratory activity rather than ROS production that plays a more central role in antibiotic tolerance.

In conclusion, a significant gap in the current literature is the lack of a comprehensive understanding of how bacterial cell metabolism undergoes changes during the transition to a tolerant state. Identifying the specific metabolic pathways that gain significance for cell survival in this context is crucial. This knowledge can pave the way for the development of more informed and targeted treatment strategies, ultimately enhancing our ability to combat tolerant cells and improve overall treatment outcomes.

## Materials and methods
### Bacterial strains and plasmids
All experiments were conducted using *E. coli* K-12 MG1655 wild-type (WT) and its derivative strains. *E. coli* K-12 MG1655, MO strains (carrying the mCherry expression system), *hipA7* (high persister strain), and pUA66 plasmids were obtained from Mark P. Brynildsen at Princeton University. *E. coli* MO strain was used to monitor cell proliferation at single cell level due to its chromosomally integrated isopropyl β-D-1-thiogalactopyranoside (IPTG)-inducible mCherry expression cassette (*Orman and Brynildsen, 2013b*; *Orman and Brynildsen, 2013a*; *Orman and Brynildsen, 2015*). *E. coli* K-12 BW25113 WT and single deletions were obtained from Dharmacon Keio Collection (Dharmacon, Catalog# OEC4988,

Lafayette, CO, USA). The mutant strains in this study were generated using the Datsenko-Wanner method (**Datsenko and Wanner, 2000**). The pUA66-EV was generated by the removal of the *gfp* gene from the plasmid. The *crp* gene with its promoter was cloned into the modified pUA66 plasmid to obtain the pUA66-*crp* expression system. The *sdhABCD* reporter (pMSs201-P*sdhABCD*-*gfp*) and the *cyaA* reporter (pMSs201-P*cyaA*-*gfp*) were obtained from a previous study (**Zaslaver et al., 2006**). The cloning method was followed according to a standard method from NEB (**Tirabassi and Bio, 2014**). Genetic modifications were verified by PCR and gene sequencing (Genewiz, South Plainfield, NJ, USA). A complete list of strains, plasmids, and oligonucleotides used in this study is presented in Appendix 1—key resources table.

## Media, chemicals, and culture conditions

All chemicals used in this study were purchased from Fisher Scientific (Atlanta, GA, USA), VWR International (Pittsburgh, PA, USA), or Sigma Aldrich (St. Louis, MO, USA). Luria-Bertani (LB) medium was prepared by combining 5 g of yeast extract, 10 g of tryptone, and 10 g of sodium chloride in 1 L of autoclaved deionized (DI) water. LB agar media was prepared by mixing 40 g of pre-mixed LB agar with 1 L of autoclaved DI water; LB agar media were used to enumerate CFUs (**Amato et al., 2013**; **Keren et al., 2004a**; **Orman and Brynildsen, 2015**). For washing cells and removing chemicals and antibiotics before plating on agar media, 1 X Phosphate-Buffered Saline (PBS) was employed. In the persister assay, concentrations of 5 μg/mL of ofloxacin (OFX), 200 μg/mL of ampicillin (AMP), and 50 μg/mL of gentamicin (GEN) were used (**Allison et al., 2011**; **De Groote et al., 2009**; **Keren et al., 2004a**; **Keren et al., 2004b**). The retention of plasmids necessitated 50 μg/mL of kanamycin (KAN) in the culture media (**Orman and Brynildsen, 2015**). Fluorescent protein expression was induced using 1 mM IPTG (**Orman and Brynildsen, 2015**). Overnight pre-cultures were prepared in 14 mL Falcon test tubes containing 2 mL of LB medium, inoculated from a 25% glycerol cell stock stored at –80 °C, and incubated for 24 hr at 37 °C with shaking at 250 revolutions per minute (rpm). Main cultures were established by diluting the overnight pre-cultures at a ratio of 1:1000 into 2 mL fresh LB medium in 14 mL Falcon test tubes. Experimental cell cultures were prepared by further dilution of the main cultures into either 25 mL fresh LB medium in 250 mL baffled flasks or 2 mL fresh LB medium in 14 mL Falcon test tubes. Cultures at t=5 hr and t=24 hr were defined as early and late stationary phase cultures, respectively. Detailed experimental procedures are outlined below.

## Cell growth and persister assays

Main cultures were prepared by diluting the overnight pre-cultures at a ratio of 1:1000 into 2 mL of fresh LB medium in 14 mL Falcon test tubes. These cultures were then incubated at 37 °C with shaking at 250 rpm. Cell growth was monitored by measuring the optical density at 600 nm wavelength ($OD_{600}$) using a Varioskan LUX Multimode Microplate Reader (Thermo Fisher, Waltham, MA, USA). The plate reader data was collected using Skanlt Software V 5.0. Cell cultures at both early and late stationary phases were collected from the test tubes and transferred to the baffled flasks to achieve ~$5 \times 10^7$ cells/mL. This concentration represents an approximately 100-fold dilution of the WT main culture into a fresh medium within the flask. It is important to highlight that we consistently employed flow cytometry to quantify the initial cell count (refer to the section 'Monitoring cell division' for comprehensive details). As needed, adjustments in cell number-to-volume were executed to ensure the same cell number among both WT and mutant strains. These cultures were then treated with antibiotics at the indicated concentrations and were cultured with shaking at 37 °C for 20 hr. After treatment, a 1 mL sample from each flask was transferred to a microcentrifuge tube for a rigorous washing protocol to minimize antibiotic carryover. The samples were centrifuged at 13,300 RPM ($17,000 \times g$) for 3 min, and >950 μL of supernatant was carefully removed without disturbing the pellet. The pellet was then resuspended in 950 μL of PBS, achieving a >20-fold dilution of residual antibiotics. This wash step was repeated once more, resulting in a cumulative >400-fold dilution. After the final wash, cells were resuspended in 100 μL of PBS. A 10 μL aliquot of this suspension was serially diluted in a 96-well round-bottom plate, and 10 μL from each dilution was spotted onto an agar plate. The remaining 90 μL of the sample was also plated to quantify CFU levels around the limit of detection, (~1 CFU/mL). Plates were incubated at 37 °C for 16 hr (or up to 48 hr) to allow CFUs to develop. When necessary, this washing procedure was repeated up to six times to assess its impact on CFU recovery.

## Mid-exponential phase persister assays

Overnight cultures of *E. coli* K-12 MG1655 or BW25113 WT, and mutant strains were diluted 1:100 in 2 mL of LB medium in 14 mL Falcon tubes and incubated at 37 °C with shaking at 250 rpm. At mid-exponential phase ($OD_{600}$~0.25), cells were challenged with the indicated antibiotics and concentrations. At designated time points, cells (1 mL) were collected, washed with 1 X PBS, and plated on agar to determine colony-forming units (CFU). To quantify the initial cell count, 10 μL of the culture before treatments was serially diluted and plated on LB agar. CFU levels were assessed after incubating the plates for at least 16 hr at 37 °C.

## cAMP profile assay

An overnight pre-culture of *E. coli* K-12 MG1655 WT was prepared in test tubes with 2 mL of LB medium. The culture was incubated at 37 °C with shaking at 250 rpm for 24 hr. For the experimental cultures, a 1:1000 dilution was made in fresh medium. At time points 0, 2, 4, 6, 8, and 24 hr, 100 μL of the experimental culture was washed with cold 1 X PBS. After centrifuging at 13,300 rpm (17,000 x *g*), the supernatant was removed. The cells were then resuspended in 100 μL of cell lysis buffer from the Cyclic AMP XP Assay Kit (Catalog# 4339 S, Cell Signaling Technology, Danvers, MA, USA) on ice for 10 min. Next, 50 μL of lysed cells was mixed with the kit's horseradish peroxidase (HRP)-linked cAMP solution in the cAMP assay plate. This mixture was incubated at room temperature for 3 hr on a plate shaker at 250 rpm. Following the 3 hr incubation, the plate content was discarded, and the plate was washed four times with the kit's Wash Buffer. Then, 100 μL of tetramethylbenzidine (TMB) substrate was added to allow color development, and after 30 min, 100 μL of the stop solution provided by the kit was added. The absorbance was measured at 450 nm using a plate reader. The standard curve was prepared using the same conditions and the standard cAMP solutions provided by the kit to determine cAMP concentrations.

## Metabolomics

Metabolites from both Δ*crp* and WT cells were analyzed at the Metabolon, Inc facility (Morrisville, NC, USA). Cells were cultured until the early stationary phase and late stationary phase at 37 °C with shaking at 250 rpm. Afterward, cells were collected through centrifugation at 4700 rpm at 37 °C for 15 min, yielding a pellet of approximately 100 μL containing around $10^{10}$ cells. Subsequently, the cells were washed once with 1 X PBS and then centrifuged (13,000 rpm, 3 min at 4 °C). Following this, they were frozen in an ethanol/dry ice bath for 10 min. The extracts from both mutant and WT cells were subjected to analysis using ultra-high-performance liquid chromatography-tandem accurate mass spectrometry (MS), a process aimed at identifying a wide range of metabolites. Sample extraction, preparation, instrument settings, and conditions for the MS platform adhered to Metabolon's protocols (as detailed in our previous study) (*Mohiuddin et al., 2022*). To identify the sample's metabolites among potential false positives from instrument noise, process artifacts, and redundant ion features, the results were cross-referenced with Metabolon's extensive metabolite library (standards). Data normalization was carried out based on protein concentration, determined using the Bradford assay. The significant difference between mutant and WT was identified using Welch's two-sample t-test. Further analysis involved pathway enrichment assessment using MetaboAnalyst (*Lu et al., 2023*). This involved inputting the upregulated and downregulated metabolites based on chosen thresholds. A comprehensive overview of metabolite measurements, pathway enrichment, statistical analyses, and data representations can be found in our previously published study (*Mohiuddin et al., 2022*).

## Proteomics

Overnight cultures for both *E. coli* K-12 MG1655 WT and Δ*crp* strains were prepared using 2 mL of LB medium. Incubation was carried out at 37 °C and 250 rpm for 24 hr. The following day, the main cultures were established under the same conditions, using a 1000-fold dilution of the overnight culture in 2 mL fresh LB medium. After 24 hours, the $OD_{600}$ of both WT and mutant strains was measured and adjusted to an $OD_{600}$ of 2.5. For further processing, 2 mL of the main culture was washed twice with cold 1 X PBS, maintaining the cold environment throughout. Centrifugation conditions were set at 4 °C, 13,000 rpm for 3 min. Before the final centrifugation, a cell count was conducted using flow cytometry. This involved using 10 μL of washed culture and 990 μL of 1 X PBS. Subsequent to centrifugation, the pellets were collected. Cell lysis was carried out using 300 μL of

NEBExpress *E. coli* Lysis Reagent (Catalog# P8116S, Ipswich, MA, USA) at room temperature for 30 mins. Following this, the lysed samples were centrifuged at 16,600 x *g* for 10 min, and 250 μL of supernatants were collected from each sample for the assay. The total protein concentration of the supernatants was determined using the bicinchoninic acid (BCA) assay (Catalog# 23225, Thermo Fisher Scientific, Waltham, MA, USA). In a 96-well plate, 25 μL of each cell lysate sample, diluted 5 and 10 times with ultra-pure DI water, were loaded into each well. Subsequently, 200 μL of the BCA working reagent (50:1, Reagent A:B) was added to each well and mixed on a plate shaker for 30 s. The plate was then incubated at 37 °C for 30 min, followed by cooling for 5 min at room temperature, shielded from light. The absorbance was finally measured at 562 nm using a plate reader, and the total protein concentration in each sample was calculated using a standard curve prepared from standard protein solutions. Protein analysis for both WT and mutant was conducted by the proteomics service at UT Health's Clinical and Translational Proteomics Service Center (Houston, TX). The samples underwent acetone precipitation, during which proteins were precipitated by exposing them to –20 °C for 3 hours. Following this, a centrifugation step at 12,000 x *g* for 5 min separated the precipitated pellets. These pellets were subsequently subjected to denaturation and reduction using a mixture containing 30 μL of 6 M urea, 20 mM DTT in 150 mM Tris HCl (pH 8.0) at 37 °C for 40 min. Afterward, alkylation was carried out with 40 mM iodoacetamide in the absence of light for 30 min. To prepare for digestion, the reaction mixture was diluted 10-fold using 50 mM Tris-HCl (pH 8.0) and then incubated overnight at 37 °C with trypsin at a 1:30 enzyme-to-substrate ratio. The digestion process was terminated by adding an equal volume of 2% formic acid, followed by desalting using Waters Oasis HLB 1 mL reverse phase cartridges, following the vendor's recommended procedure. Finally, the elutes were dried using vacuum centrifugation. Approximately 1 μg of the tryptic digest, prepared in a solution containing 2% acetonitrile and 0.1% formic acid in water, underwent analysis using LC/MS/MS. The instrument used was the Orbitrap Fusion Tribrid mass spectrometer by Thermo Fisher Scientific, connected to a Dionex UltiMate 3000 Binary RSLCnano System. The separation of peptides occurred on an analytical C18 column with dimensions of 100 μm ID x 25 cm, featuring 5 μm particles and an 18 Å pore size. Peptides were eluted at a flow rate of 350 nL/min. The gradient conditions applied were as follows: a gradient starting from 3% B and increasing to 22% B over a duration of 90 min, followed by a step to 22–35% B for 10 min, then another step to 35–90% B for 10 min, and finally, maintaining 90% B for an additional 10 min (Solvent A was composed of 0.1% formic acid in water, while solvent B contained 0.1% formic acid in acetonitrile). The peptides were analyzed using a data-dependent acquisition method. The Orbitrap Fusion MS operated by measuring FTMS1 spectra with a resolution of 120,000 FWHM, scanning in the m/z range of 350–1500, using an AGC target set to 2E5, and with a maximum injection time of 50ms. Within a maximum cycle time of 3 s, ITMS2 spectra were collected in rapid scan mode. High Collision Dissociation (HCD) was employed with a normalized collision energy (NCE) of 34, an isolation window of 1.6 m/z, an AGC target set to 1E4, and a maximum injection time of 35ms. Dynamic exclusion was implemented for a duration of 35 s to prevent repeated analysis of the same ions. For the experimental analysis, the Thermo Fisher Scientific Proteome Discoverer software version 1.4 was utilized to process the raw data files. The spectra were subjected to analysis against the *E. coli* proteome database (Swiss-Prot 29,161) through the Sequest HT search engine. Additionally, the spectra were compared against a decoy database, employing a target false discovery rate (FDR) of 1% for stringent criteria and 5% for more relaxed criteria. The enzymatic cleavage allowance for trypsin included up to two potential missed cleavages. The MS tolerance was defined as 10 ppm, while the MS/MS tolerance was set at 0.6 Da. Fixed modification involved carbamidomethylation on cysteine residues, and variable modifications encompassed methionine oxidation and asparagine deamidation. For proteomics data processing and fold change calculations, the approaches were essentially followed by the method paper from *Aguilan et al., 2020*. Then, the STRING tool V12.0 (*Szklarczyk et al., 2021*; *Szklarczyk et al., 2023*) was employed to find the significant networks among input proteins. To generate the protein network and pathway enrichment analysis, we input the protein identifiers (Accession numbers) for upregulated or downregulated proteins with at least a twofold increase or reduction, respectively. *E. coli* K-12 was selected as the organism of interest. We opted for evidence as the criterion for network edges, prioritizing the type of interaction evidence, helping us conduct an automated pathway-enrichment analysis, centering on the entered proteins and identifying pathways that occurred more frequently than expected. This analysis was grounded in the statistical background of the entire genome and encompasses various

functional pathway classification frameworks, such as Gene Ontology annotations, KEGG pathways, and Uniprot keywords, as detailed elsewhere (*Szklarczyk et al., 2021*; *Szklarczyk et al., 2023*). In pathway enrichment analysis, the 'strength score', calculated as $Log_{10}$(observed/expected), serves to assess the degree or significance of enrichment within a specific biological pathway. This metric reflects the magnitude of the enrichment effect, with a higher score indicating stronger enrichment. The score is derived from the ratio of (i) annotated proteins in the network for a given term to (ii) the expected number of proteins annotated with the same term in a random network of equivalent size (*Szklarczyk et al., 2021*; *Szklarczyk et al., 2023*). To gauge the significance of enrichment, False Discovery Rate (FDR) is employed. FDR scores represent p-values corrected for multiple testing within each category using the Benjamini–Hochberg procedure (*Szklarczyk et al., 2021*; *Szklarczyk et al., 2023*).

## Monitoring cell division

To monitor cell division and quantify non-growing cells, we utilized inducible fluorescent protein (mCherry) expression. Overnight pre-cultures of *E. coli* MO were prepared with 2 mL of LB medium containing 1 mM of IPTG. These cultures were grown in test tubes at 37 °C with shaking at 250 rpm for 24 hr. Main cultures were established by diluting the overnight pre-cultures (at a ratio of 1:1000) into 2 mL of fresh LB medium in 14 mL Falcon test tubes. These cultures were incubated at 37 °C with shaking at 250 rpm. Cells were allowed to grow until they reached the early stationary phase and the late stationary phase. The mCherry-positive cells were then collected, washed twice with 1 X PBS to remove the IPTG from the culture, and subsequently re-suspended in fresh 2 ml LB media in test tubes to achieve ~$5 \times 10^7$ cells/mL. This concentration represents an approximately 100-fold dilution. When needed, adjustments in cell number-to-volume were made to ensure the same cell number among both WT and mutant strains. In the experimental culture test tubes, 2 mL of LB medium was added, and the volume of washed cells was inoculated to achieve an $OD_{600}$ of 0.0286. The culture was then incubated at 37 °C with shaking at 250 rpm. At specific time points (0, 1, 2, and 2.5 hr), cells were collected and re-suspended in 1 X PBS to measure their fluorescent protein content using flow cytometry. For flow cytometry analysis, cells were collected and diluted to a desired cell density (~$10^6$–$10^7$ cells/mL) in 1 mL of 1 X PBS in flow cytometry tubes (5 mL round-bottom Falcon tubes, size: 12x75 mm). The flow cytometry analysis was conducted using a NovoCyte 3000RYB instrument (ACEA Bioscience Inc, San Diego, CA, USA). During flow cytometry analysis, a slow sample flow rate of 14 µL/min was chosen, along with a sample stream diameter (core diameter) of 7.7 µm. The instrument maintained a constant sheath flow rate of 6.5 mL/min. The core diameter was calculated using the ratio of the sample flow rate to the sheath flow rate. These specific conditions were selected to achieve improved data resolution for the size of *E. coli* cells. Flow diagrams utilized forward and side scatter signals from viable cells, alongside a control of solvent devoid of cells, to ascertain the presence of cells. For the flow cytometry analysis, cells were excited at a 561 nm wavelength, and the red fluorescence was detected using a 615/20 nm bandpass filter.

## Flow cytometry analysis of cell growth

Overnight cultures of *E. coli* K-12 MG1655 MO WT and mutant strains were diluted at a ratio of 1:1000 into 2 mL of fresh LB medium, placed in 14 mL Falcon test tubes, and incubated at 37 °C with shaking at 250 rpm for 24 hr. For the main cultures, a similar strategy was employed. The cultures were diluted at a ratio of 1:100 into 2 mL of fresh LB medium in 14 mL Falcon test tubes. These cultures were then incubated at 37 °C with shaking at 250 rpm. At specific time points, including t=0, 20 min, 40 min, and 1–5 hr, the cell growth was halted. This was achieved by diluting the cells in 1 X PBS containing 25 µg/mL of chloramphenicol (CAM). The CAM treatment allowed for subsequent analysis without further division. Flow cytometry was then utilized to measure the number of cells at each of these time points. This approach provided insight into cell division dynamics and allowed for the quantification of cell populations under specific conditions.

## Fluorescent protein expression assay for reporter genes

Mutant and control strains were derived from *E. coli* K-12 MG1655 and carried pMSs201-*gfp* plasmids incorporating $P_{sdhABCD}$ or $P_{cyaA}$ gene promoters. Overnight pre-cultures were prepared using 2 mL of LB medium supplemented with 50 µg/mL KAN. These cultures were incubated in test tubes

within a shaker at 37 °C for 24 hr. Main cultures were established by diluting the overnight pre-cultures at a ratio of 1:1000 into 2 mL of fresh LB medium within 14 mL Falcon test tubes. These main cultures were maintained at 37 °C with shaking at 250 rpm. Cell cultures at the desired growth phase were collected and then diluted to attain a desired cell density of around $10^6$–$10^7$ cells/mL in 1 mL of 1 X PBS within flow cytometry tubes. This allowed for subsequent flow cytometry analysis, using the same conditions as described earlier for monitoring cell division (refer to 'Monitoring cell division'). During analysis, a laser emitting light at 488 nm was used to excite the cells, and the resulting green fluorescence was detected using a 530/30 nm bandpass filter. This setup enabled the examination of the fluorescence patterns of the cells, offering insights into their dynamics under different conditions.

## Redox Sensor Green assay

To gauge bacterial metabolic activity, we employed the Redox Sensor Green (RSG) dye from Thermo Fisher (Catalog# B34954, Thermo Fisher Scientific, Waltham, MA). *E. coli* K-12 MG1655 WT and mutant cells from the desired growth phase were diluted at a ratio of 1:100 in 1 mL of 1 X PBS. To this solution, 1 µL of the RSG dye was added to flow cytometry tubes. After a brief vortexing to ensure uniform mixing, the samples were incubated at 37 °C in darkness for 10 min. Subsequently, these samples were subjected to flow cytometry analysis. For the flow cytometry analysis, the same methodology as employed in 'Monitoring cell division' was followed, with one variation. Cells were excited at 488 nm during analysis, and the resulting green fluorescence was detected using a 530/30 nm bandpass filter. This setup allowed us to assess the fluorescence patterns, reflecting the metabolic activity of the bacterial cells under different conditions. As a control measure, cells were treated with 20 µM of carbonyl cyanide m-chlorophenyl hydrazone (CCCP) for 5 min before the addition of the RSG dye. This served to validate the assay's sensitivity to changes in metabolic activity (*Figure 5—figure supplement 1*).

## Metabolic activity of non-lysing cells and VBNC cell quantification

Overnight cultures of *E. coli* K-12 MG1655 MO WT and mutant strains were diluted at a ratio of 1:1000 into 2 mL of fresh LB medium supplemented with 1 mM IPTG. These cultures were established in 14 mL Falcon test tubes and incubated at 37 °C with shaking at 250 rpm for 24 hr. Treatment cultures were prepared by diluting the main cultures at a ratio of 1:100 into 25 mL of fresh LB medium supplemented with 1 mM IPTG. These cultures were set up in 250 mL baffled flasks and contained 200 µg/mL AMP. They were then cultured at 37 °C with shaking at 250 rpm for 20 hr. Both before and after the treatment, 1 mL samples were collected from the cultures. These samples were subjected to a washing procedure with 1 X PBS to eliminate the antibiotic present in the samples. The washed cells were then resuspended in 1 mL of 1 X PBS within flow cytometry tubes. To measure the metabolic activity of the non-lysing cells, the RSG dye was employed as described above. Intact cells (following antibiotic treatment), stained as live with RSG, comprised both persister and VBNC cells. Persister levels were quantified by plating the cells on an agar medium, as described previously (*Orman and Brynildsen, 2013b*). As VBNC cells cannot grow on agar medium, their enumeration involved subtracting the number of persister cells from the total number of intact cells.

## Screening *E. coli* (K-12 BW25113) Keio knockout collection

Overnight cultures of individual mutant strains, along with their parental strain K-12 BW25113 WT harboring a kanamycin-resistant marker, were diluted at a ratio of 1:1000 in fresh LB medium containing 50 µg/mL of KAN. This was done in 14 mL Falcon test tubes and the cultures were then incubated at 37 °C with shaking at 250 rpm. Upon reaching the late stationary phase, cells were further diluted at a ratio of 1:100 in fresh medium supplemented with antibiotics at specified concentrations. These cultures were once again incubated at 37 °C with shaking for 20 hr. Following the 20 hr treatment period, the same methodology described in the section 'Cell growth and persister assays' was employed to quantify the number of persisters. This approach allowed for an assessment of the impact of antibiotics on the formation of persister cells for both mutant strains and the parental K-12 BW25113 WT strain.

## Persister quantitation in *E. coli* K-12 MG1655 single gene deletions

Overnight cultures of mutant strains were diluted at a ratio of 1:1000 in 14 mL Falcon test tubes containing 2 mL of LB medium. These cultures were then incubated at 37 °C with shaking at 250 rpm. Upon reaching the late stationary phase, cells were diluted at a ratio of 1:100 in fresh medium supplemented with antibiotics at specified concentrations. The cultures were once again subjected to shaking at 37 °C for 20 hr. The same method described earlier, referred to as 'Cell growth and persister assays', was employed to quantify the number of persister cells resulting from this treatment. This approach allowed for the assessment of the impact of antibiotic exposure on persister cell formation within the mutant strains.

## Statistics and reproducibility

A nonlinear logarithmic model was employed to create biphasic kill curves (*Mohiuddin et al., 2020a*; *Windels et al., 2019*). The significance of these kill curves was determined through the utilization of F-statistics (*Mohiuddin et al., 2020a*; *Windels et al., 2019*). Metabolomics data were subjected to analysis using Welch's two-sample t-test in order to identify metabolites that significantly differed between the control and mutant groups (*Yuen, 1974*). For all experiments, a minimum of three independent biological replicates were conducted, unless explicitly stated otherwise. In each figure (excluding flow diagrams), the data for each time point are represented as the mean value accompanied by the standard deviation. In terms of statistical significance analysis, the designated threshold values for p were set as follows: $*p<0.05$, $**p<0.01$, $***p<0.001$, and $****p<0.0001$. All figures were generated using GraphPad Prism 10.3.0. The statistical analyses were carried out using the statistical functions of GraphPad Prism 10.3.0. For the clustering of metabolomics and proteomics data, the 'Clustergram' function of MATLAB (V R2020b) was employed. FlowJo (V 10.8.1) was the tool used to analyze the data acquired from flow cytometry.

## Acknowledgements

We would like to express our gratitude to the members of Dr. Orman's laboratory for their support and assistance throughout this study. We also thank Metabolon, Inc for their expertise in metabolomics experiments and data analysis, as well as UT Health's Clinical and Translational Proteomics Service Center for their contribution to the proteomics experiments.

## Additional information

### Funding

| Funder | Grant reference number | Author |
| --- | --- | --- |
| National Science Foundation | 2044375 | Mehmet Orman |
| National Institutes of Health | R01AI143643 | Mehmet Orman |

The funders had no role in study design, data collection and interpretation, or the decision to submit the work for publication.

### Author contributions

Han G Ngo, Conceptualization, Data curation, Formal analysis, Validation, Investigation, Methodology, Writing – original draft, Writing – review and editing; Sayed Golam Mohiuddin, Data curation, Formal analysis, Methodology, Writing – original draft, Writing – review and editing; Aina Ananda, Software, Formal analysis, Validation, Methodology; Mehmet Orman, Conceptualization, Formal analysis, Supervision, Funding acquisition, Investigation, Methodology, Writing – original draft, Project administration, Writing – review and editing

### Author ORCIDs

Han G Ngo https://orcid.org/0000-0002-3860-4550
Sayed Golam Mohiuddin https://orcid.org/0000-0001-7613-6324

Mehmet Orman [ORCID] https://orcid.org/0000-0001-8499-9154

Reviewer #1 (Public review): https://doi.org/10.7554/eLife.99735.3.sa1
Author response https://doi.org/10.7554/eLife.99735.3.sa2

## Additional files

### Supplementary files

Supplementary file 1. MIC of antibiotics and concentrations of bactericidal antibiotics used in persister assays.

Supplementary file 2. Analyzed metabolomics data.

Supplementary file 3. Analyzed proteomics data.

Supplementary file 4. Analyzed persister survival fraction data.

Supplementary file 5. The knockout strains generated using the *E. coli* K-12 MG1655 background in this study.

Supplementary file 6. Persister levels of *E. coli* K-12 MG1655 WT, Δ*crp*, and Δ*cyaA* strains in late stationary phase.

Supplementary file 7. Persister levels of *E. coli* K-12 MG1655 (A) and BW25113 (B) WT, Δ*crp*, and Δ*cyaA* strains in the exponential growth phase.

MDAR checklist

### Data availability

All data generated or analyzed during this study are included in the manuscript and supporting files.

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

# Appendix 1

**Appendix 1—key resources table**

| Reagent type (species) or resource | Designation | Source or reference | Identifiers | Additional information |
|---|---|---|---|---|
| Strain, strain background (*Escherichia coli*) | K-12 MG1655 Wild Type | Gift from Dr. Mark P. Brynildsen | | |
| Strain, strain background (*E. coli*) | K-12 MG1655 hipA7 | Gift from Dr. Mark P. Brynildsen | | |
| Strain, strain background (*E. coli*) | K-12 MG1655 hipA7Δcrp | This study | | |
| Strain, strain background (*E. coli*) | K-12 MG1655 hipA7ΔcyaA | This study | | |
| Strain, strain background (*E. coli*) | K-12 BW25113 Wild Type | Keio collection | Catalog # OEC5042 | |
| Strain, strain background (*E. coli*) | K-12 MG1655 MO | Gift from Dr. Mark P. Brynildsen | | |
| Strain, strain background (*E. coli*) | K-12 MG1655 MO Δcrp | This study | | |
| Strain, strain background (*E. coli*) | K-12 MG1655 MO ΔcyaA | This study | | |
| Strain, strain background (*E. coli*) | K-12 MG1655 Δcrp | This study | | |
| Strain, strain background (*E. coli*) | K-12 MG1655 ΔcyaA | This study | | |
| Strain, strain background (*E. coli*) | K-12 MG1655 ΔsucA | This study | | |
| Strain, strain background (*E. coli*) | K-12 MG1655 Δlpd | This study | | |
| Strain, strain background (*E. coli*) | K-12 MG1655 ΔsucC | This study | | |
| Strain, strain background (*E. coli*) | K-12 MG1655 ΔsdhA | This study | | |
| Strain, strain background (*E. coli*) | K-12 MG1655 ΔgltA | This study | | |
| Strain, strain background (*E. coli*) | K-12 MG1655 ΔaceE | This study | | |
| Strain, strain background (*E. coli*) | K-12 MG1655 ΔtktB | This study | | |
| Strain, strain background (*E. coli*) | K-12 MG1655 Δmdh | This study | | |
| Strain, strain background (*E. coli*) | K-12 MG1655 ΔnuoI | This study | | |
| Strain, strain background (*E. coli*) | K-12 MG1655 ΔnuoM | This study | | |
| Strain, strain background (*E. coli*) | K-12 MG1655 ΔatpA | This study | | |
| Strain, strain background (*E. coli*) | K-12 MG1655 ΔatpB | This study | | |
| Strain, strain background (*E. coli*) | K-12 MG1655 ΔatpC | This study | | |

*Appendix 1 Continued on next page*

*Appendix 1 Continued*

| Reagent type (species) or resource | Designation | Source or reference | Identifiers | Additional information |
|---|---|---|---|---|
| Strain, strain background *(E. coli)* | K-12 MG1655 Δ*atpD* | This study | | |
| Strain, strain background *(E. coli)* | K-12 MG1655 Δ*frdC* | This study | | |
| Strain, strain background *(E. coli)* | K-12 MG1655 Δ*pgi* | This study | | |
| Strain, strain background *(E. coli)* | K-12 MG1655 Δ*zwf* | This study | | |
| Strain, strain background *(E. coli)* | K-12 MG1655 Δ*talA* | This study | | |
| Strain, strain background *(E. coli)* | K-12 BW25113 Δ*acnB* | Keio collection | Catalog # OEC4988 | |
| Strain, strain background *(E. coli)* | K-12 BW25113 Δ*sucA* | Keio collection | Catalog # OEC4988 | |
| Strain, strain background *(E. coli)* | K-12 BW25113 Δ*sucB* | Keio collection | Catalog # OEC4988 | |
| Strain, strain background *(E. coli)* | K-12 BW25113 Δ*sucC* | Keio collection | Catalog # OEC4988 | |
| Strain, strain background *(E. coli)* | K-12 BW25113 Δ*sdhD* | Keio collection | Catalog # OEC4988 | |
| Strain, strain background *(E. coli)* | K-12 BW25113 Δ*sdhC* | Keio collection | Catalog # OEC4988 | |
| Strain, strain background *(E. coli)* | K-12 BW25113 Δ*sdhB* | Keio collection | Catalog # OEC4988 | |
| Strain, strain background *(E. coli)* | K-12 BW25113 Δ*sdhA* | Keio collection | Catalog # OEC4988 | |
| Strain, strain background *(E. coli)* | K-12 BW25113 Δ*aceF* | Keio collection | Catalog # OEC4988 | |
| Strain, strain background *(E. coli)* | K-12 BW25113 Δ*talB* | Keio collection | Catalog # OEC4988 | |
| Strain, strain background *(E. coli)* | K-12 BW25113 Δ*cyoA* | Keio collection | Catalog # OEC4988 | |
| Strain, strain background *(E. coli)* | K-12 BW25113 Δ*cyoB* | Keio collection | Catalog # OEC4988 | |
| Strain, strain background *(E. coli)* | K-12 BW25113 Δ*cyoC* | Keio collection | Catalog # OEC4988 | |
| Strain, strain background *(E. coli)* | K-12 BW25113 Δ*cyoD* | Keio collection | Catalog # OEC4988 | |
| Strain, strain background *(E. coli)* | K-12 BW25113 Δ*acnA* | Keio collection | Catalog # OEC4988 | |
| Strain, strain background *(E. coli)* | K-12 BW25113 Δ*icd* | Keio collection | Catalog # OEC4988 | |
| Strain, strain background *(E. coli)* | K-12 BW25113 Δ*fumA* | Keio collection | Catalog # OEC4988 | |
| Strain, strain background *(E. coli)* | K-12 BW25113 Δ*fumB* | Keio collection | Catalog # OEC4988 | |
| Strain, strain background *(E. coli)* | K-12 BW25113 Δ*fumC* | Keio collection | Catalog # OEC4988 | |

*Appendix 1 Continued on next page*

*Appendix 1 Continued*

| Reagent type (species) or resource | Designation | Source or reference | Identifiers | Additional information |
|---|---|---|---|---|
| Strain, strain background (*E. coli*) | K-12 BW25113 Δmdh | Keio collection | Catalog # OEC4988 | |
| Strain, strain background (*E. coli*) | K-12 BW25113 Δpgi | Keio collection | Catalog # OEC4988 | |
| Strain, strain background (*E. coli*) | K-12 BW25113 ΔpfkA | Keio collection | Catalog # OEC4988 | |
| Strain, strain background (*E. coli*) | K-12 BW25113 ΔtpiA | Keio collection | Catalog # OEC4988 | |
| Strain, strain background (*E. coli*) | K-12 BW25113 ΔgpmM | Keio collection | Catalog # OEC4988 | |
| Strain, strain background (*E. coli*) | K-12 BW25113 ΔppsA | Keio collection | Catalog # OEC4988 | |
| Strain, strain background (*E. coli*) | K-12 BW25113 ΔpykF | Keio collection | Catalog # OEC4988 | |
| Strain, strain background (*E. coli*) | K-12 BW25113 ΔpykA | Keio collection | Catalog # OEC4988 | |
| Strain, strain background (*E. coli*) | K-12 BW25113 ΔmaeB | Keio collection | Catalog # OEC4988 | |
| Strain, strain background (*E. coli*) | K-12 BW25113 Δpck | Keio collection | Catalog # OEC4988 | |
| Strain, strain background (*E. coli*) | K-12 BW25113 ΔrpiB | Keio collection | Catalog # OEC4988 | |
| Strain, strain background (*E. coli*) | K-12 BW25113 Δrpe | Keio collection | Catalog # OEC4988 | |
| Strain, strain background (*E. coli*) | K-12 BW25113 ΔtktB | Keio collection | Catalog # OEC4988 | |
| Strain, strain background (*E. coli*) | K-12 BW25113 ΔtalA | Keio collection | Catalog # OEC4988 | |
| Strain, strain background (*E. coli*) | K-12 BW25113 Δzwf | Keio collection | Catalog # OEC4988 | |
| Strain, strain background (*E. coli*) | K-12 BW25113 Δgnd | Keio collection | Catalog # OEC4988 | |
| Strain, strain background (*E. coli*) | K-12 BW25113 Δppc | Keio collection | Catalog # OEC4988 | |
| Strain, strain background (*E. coli*) | K-12 BW25113 ΔfrdA | Keio collection | Catalog # OEC4988 | |
| Strain, strain background (*E. coli*) | K-12 BW25113 ΔfrdB | Keio collection | Catalog # OEC4988 | |
| Strain, strain background (*E. coli*) | K-12 BW25113 ΔfrdC | Keio collection | Catalog # OEC4988 | |
| Strain, strain background (*E. coli*) | K-12 BW25113 ΔfrdD | Keio collection | Catalog # OEC4988 | |
| Strain, strain background (*E. coli*) | K-12 BW25113 ΔadhE | Keio collection | Catalog # OEC4988 | |
| Strain, strain background (*E. coli*) | K-12 BW25113 ΔpflB | Keio collection | Catalog # OEC4988 | |
| Strain, strain background (*E. coli*) | K-12 BW25113 ΔaceB | Keio collection | Catalog # OEC4988 | |

*Appendix 1 Continued on next page*

*Appendix 1 Continued*

| Reagent type (species) or resource | Designation | Source or reference | Identifiers | Additional information |
|---|---|---|---|---|
| Strain, strain background (*E. coli*) | K-12 BW25113 Δ*aceA* | Keio collection | Catalog # OEC4988 | |
| Strain, strain background (*E. coli*) | K-12 BW25113 Δ*glcB* | Keio collection | Catalog # OEC4988 | |
| Strain, strain background (*E. coli*) | K-12 BW25113 Δ*nuoA* | Keio collection | Catalog # OEC4988 | |
| Strain, strain background (*E. coli*) | K-12 BW25113 Δ*nuoH* | Keio collection | Catalog # OEC4988 | |
| Strain, strain background (*E. coli*) | K-12 BW25113 Δ*nuoJ* | Keio collection | Catalog # OEC4988 | |
| Strain, strain background (*E. coli*) | K-12 BW25113 Δ*nuoK* | Keio collection | Catalog # OEC4988 | |
| Strain, strain background (*E. coli*) | K-12 BW25113 Δ*nuoL* | Keio collection | Catalog # OEC4988 | |
| Strain, strain background (*E. coli*) | K-12 BW25113 Δ*nuoM* | Keio collection | Catalog # OEC4988 | |
| Strain, strain background (*E. coli*) | K-12 BW25113 Δ*nuoN* | Keio collection | Catalog # OEC4988 | |
| Strain, strain background (*E. coli*) | K-12 BW25113 Δ*nuoB* | Keio collection | Catalog # OEC4988 | |
| Strain, strain background (*E. coli*) | K-12 BW25113 Δ*nuoE* | Keio collection | Catalog # OEC4988 | |
| Strain, strain background (*E. coli*) | K-12 BW25113 Δ*nuoF* | Keio collection | Catalog # OEC4988 | |
| Strain, strain background (*E. coli*) | K-12 BW25113 Δ*nuoG* | Keio collection | Catalog # OEC4988 | |
| Strain, strain background (*E. coli*) | K-12 BW25113 Δ*nuoI* | Keio collection | Catalog # OEC4988 | |
| Strain, strain background (*E. coli*) | K-12 BW25113 Δ*appC* | Keio collection | Catalog # OEC4988 | |
| Strain, strain background (*E. coli*) | K-12 BW25113 Δ*appB* | Keio collection | Catalog # OEC4988 | |
| Strain, strain background (*E. coli*) | K-12 BW25113 Δ*gltA* | Keio collection | Catalog # OEC4988 | |
| Strain, strain background (*E. coli*) | K-12 BW25113 Δ*lpd* | Keio collection | Catalog # OEC4988 | |
| Strain, strain background (*E. coli*) | K-12 BW25113 Δ*fbp* | Keio collection | Catalog # OEC4988 | |
| Strain, strain background (*E. coli*) | K-12 BW25113 Δ*aceE* | Keio collection | Catalog # OEC4988 | |
| Strain, strain background (*E. coli*) | K-12 BW25113 Δ*tktA* | Keio collection | Catalog # OEC4988 | |
| Strain, strain background (*E. coli*) | K-12 BW25113 Δ*pta* | Keio collection | Catalog # OEC4988 | |
| Strain, strain background (*E. coli*) | K-12 BW25113 Δ*ackA* | Keio collection | Catalog # OEC4988 | |
| Strain, strain background (*E. coli*) | K-12 BW25113 Δ*ldhA* | Keio collection | Catalog # OEC4988 | |

*Appendix 1 Continued on next page*

*Appendix 1 Continued*

| Reagent type (species) or resource | Designation | Source or reference | Identifiers | Additional information |
|---|---|---|---|---|
| Strain, strain background (*E. coli*) | K-12 BW25113 Δ*dld* | Keio collection | Catalog # OEC4988 | |
| Strain, strain background (*E. coli*) | K-12 BW25113 Δ*cydB* | Keio collection | Catalog # OEC4988 | |
| Strain, strain background (*E. coli*) | K-12 BW25113 Δ*poxB* | Keio collection | Catalog # OEC4988 | |
| Strain, strain background (*E. coli*) | K-12 BW25113 Δ*glpX* | Keio collection | Catalog # OEC4988 | |
| Strain, strain background (*E. coli*) | K-12 BW25113 Δ*ybhA* | Keio collection | Catalog # OEC4988 | |
| Strain, strain background (*E. coli*) | K-12 BW25113 Δ*cydX* | Keio collection | Catalog # OEC4988 | |
| Strain, strain background (*E. coli*) | K-12 BW25113 Δ*fumE* | Keio collection | Catalog # OEC4988 | |
| Strain, strain background (*E. coli*) | K-12 BW25113 Δ*yggF* | Keio collection | Catalog # OEC4988 | |
| Strain, strain background (*E. coli*) | K-12 BW25113 Δ*ccp* | Keio collection | Catalog # OEC4988 | |
| Strain, strain background (*E. coli*) | K-12 BW25113 Δ*yieF* | Keio collection | Catalog # OEC4988 | |
| Strain, strain background (*E. coli*) | K-12 BW25113 Δ*wrbA* | Keio collection | Catalog # OEC4988 | |
| Strain, strain background (*E. coli*) | K-12 BW25113 Δ*atpC* | Keio collection | Catalog # OEC4988 | |
| Strain, strain background (*E. coli*) | K-12 BW25113 Δ*atpD* | Keio collection | Catalog # OEC4988 | |
| Strain, strain background (*E. coli*) | K-12 BW25113 Δ*fdhF* | Keio collection | Catalog # OEC4988 | |
| Strain, strain background (*E. coli*) | K-12 BW25113 Δ*nrfD* | Keio collection | Catalog # OEC4988 | |
| Strain, strain background (*E. coli*) | K-12 BW25113 Δ*nrfC* | Keio collection | Catalog # OEC4988 | |
| Strain, strain background (*E. coli*) | K-12 BW25113 Δ*nrfA* | Keio collection | Catalog # OEC4988 | |
| Strain, strain background (*E. coli*) | K-12 BW25113 Δ*putA* | Keio collection | Catalog # OEC4988 | |
| Strain, strain background (*E. coli*) | K-12 BW25113 Δ*dmsC* | Keio collection | Catalog # OEC4988 | |
| Strain, strain background (*E. coli*) | K-12 BW25113 Δ*torC* | Keio collection | Catalog # OEC4988 | |
| Strain, strain background (*E. coli*) | K-12 BW25113 Δ*torA* | Keio collection | Catalog # OEC4988 | |
| Strain, strain background (*E. coli*) | K-12 BW25113 Δ*hyaA* | Keio collection | Catalog # OEC4988 | |
| Strain, strain background (*E. coli*) | K-12 BW25113 Δ*hyaB* | Keio collection | Catalog # OEC4988 | |
| Strain, strain background (*E. coli*) | K-12 BW25113 Δ*hyaC* | Keio collection | Catalog # OEC4988 | |

*Appendix 1 Continued*

| Reagent type (species) or resource | Designation | Source or reference | Identifiers | Additional information |
|---|---|---|---|---|
| Strain, strain background (E. coli) | K-12 BW25113 ΔkefF | Keio collection | Catalog # OEC4988 | |
| Strain, strain background (E. coli) | K-12 BW25113 ΔnarV | Keio collection | Catalog # OEC4988 | |
| Strain, strain background (E. coli) | K-12 BW25113 ΔnarI | Keio collection | Catalog # OEC4988 | |
| Strain, strain background (E. coli) | K-12 BW25113 ΔkduI | Keio collection | Catalog # OEC4988 | |
| Strain, strain background (E. coli) | K-12 BW25113 ΔeutE | Keio collection | Catalog # OEC4988 | |
| Strain, strain background (E. coli) | K-12 BW25113 ΔhycE | Keio collection | Catalog # OEC4988 | |
| Strain, strain background (E. coli) | K-12 BW25113 ΔhycG | Keio collection | Catalog # OEC4988 | |
| Strain, strain background (E. coli) | K-12 BW25113 Δedd | Keio collection | Catalog # OEC4988 | |
| Strain, strain background (E. coli) | K-12 BW25113 Δeda | Keio collection | Catalog # OEC4988 | |
| Strain, strain background (E. coli) | K-12 BW25113 ΔfdnG | Keio collection | Catalog # OEC4988 | |
| Strain, strain background (E. coli) | K-12 BW25113 ΔfdnI | Keio collection | Catalog # OEC4988 | |
| Strain, strain background (E. coli) | K-12 BW25113 ΔglpD | Keio collection | Catalog # OEC4988 | |
| Strain, strain background (E. coli) | K-12 BW25113 ΔglpA | Keio collection | Catalog # OEC4988 | |
| Strain, strain background (E. coli) | K-12 BW25113 ΔglpB | Keio collection | Catalog # OEC4988 | |
| Strain, strain background (E. coli) | K-12 BW25113 ΔglpC | Keio collection | Catalog # OEC4988 | |
| Strain, strain background (E. coli) | K-12 BW25113 ΔhybO | Keio collection | Catalog # OEC4988 | |
| Strain, strain background (E. coli) | K-12 BW25113 ΔhybC | Keio collection | Catalog # OEC4988 | |
| Strain, strain background (E. coli) | K-12 BW25113 ΔnarY | Keio collection | Catalog # OEC4988 | |
| Strain, strain background (E. coli) | K-12 BW25113 ΔnarZ | Keio collection | Catalog # OEC4988 | |
| Strain, strain background (E. coli) | K-12 BW25113 ΔnapG | Keio collection | Catalog # OEC4988 | |
| Strain, strain background (E. coli) | K-12 BW25113 ΔnapH | Keio collection | Catalog # OEC4988 | |
| Strain, strain background (E. coli) | K-12 BW25113 ΔnapA | Keio collection | Catalog # OEC4988 | |
| Strain, strain background (E. coli) | K-12 BW25113 ΔatpA | Keio collection | Catalog # OEC4988 | |
| Strain, strain background (E. coli) | K-12 BW25113 ΔpurT | Keio collection | Catalog # OEC4988 | |

*Appendix 1 Continued on next page*

*Appendix 1 Continued*

| Reagent type (species) or resource | Designation | Source or reference | Identifiers | Additional information |
|---|---|---|---|---|
| Strain, strain background (*E. coli*) | K-12 BW25113 ΔfdoG | Keio collection | Catalog # OEC4988 | |
| Strain, strain background (*E. coli*) | K-12 BW25113 ΔfdoI | Keio collection | Catalog # OEC4988 | |
| Strain, strain background (*E. coli*) | K-12 BW25113 ΔatpB | Keio collection | Catalog # OEC4988 | |
| Strain, strain background (*E. coli*) | K-12 BW25113 ΔatpE | Keio collection | Catalog # OEC4988 | |
| Strain, strain background (*E. coli*) | K-12 BW25113 ΔatpF | Keio collection | Catalog # OEC4988 | |
| Strain, strain background (*E. coli*) | K-12 BW25113 ΔatpH | Keio collection | Catalog # OEC4988 | |
| Strain, strain background (*E. coli*) | K-12 BW25113 ΔphoA | Keio collection | Catalog # OEC4988 | |
| Strain, strain background (*E. coli*) | K-12 BW25113 ΔadhP | Keio collection | Catalog # OEC4988 | |
| Strain, strain background (*E. coli*) | K-12 BW25113 ΔeutD | Keio collection | Catalog # OEC4988 | |
| Strain, strain background (*E. coli*) | K-12 BW25113 ΔdmsA | Keio collection | Catalog # OEC4988 | |
| Strain, strain background (*E. coli*) | K-12 BW25113 ΔhybB | Keio collection | Catalog # OEC4988 | |
| Strain, strain background (*E. coli*) | K-12 BW25113 ΔnapB | Keio collection | Catalog # OEC4988 | |
| Strain, strain background (*E. coli*) | K-12 BW25113 ΔatpI | Keio collection | Catalog # OEC4988 | |
| Strain, strain background (*E. coli*) | K-12 BW25113 ΔmaeA | Keio collection | Catalog # OEC4988 | |
| Strain, strain background (*E. coli*) | K-12 BW25113 ΔpfkB | Keio collection | Catalog # OEC4988 | |
| Strain, strain background (*E. coli*) | K-12 BW25113 ΔfbaB | Keio collection | Catalog # OEC4988 | |
| Strain, strain background (*E. coli*) | K-12 BW25113 ΔrpiA | Keio collection | Catalog # OEC4988 | |
| Strain, strain background (*E. coli*) | K-12 BW25113 ΔfumD | Keio collection | Catalog # OEC4988 | |
| Strain, strain background (*E. coli*) | K-12 BW25113 ΔsucD | Keio collection | Catalog # OEC4988 | |
| Strain, strain background (*E. coli*) | K-12 BW25113 ΔfdnH | Keio collection | Catalog # OEC4988 | |
| Strain, strain background (*E. coli*) | K-12 BW25113 ΔdmsB | Keio collection | Catalog # OEC4988 | |
| Strain, strain background (*E. coli*) | K-12 BW25113 Δndh | Keio collection | Catalog # OEC4988 | |
| Strain, strain background (*E. coli*) | K-12 BW25113 ΔnarG | Keio collection | Catalog # OEC4988 | |
| Strain, strain background (*E. coli*) | K-12 BW25113 ΔnarH | Keio collection | Catalog # OEC4988 | |

*Appendix 1 Continued on next page*

*Appendix 1 Continued*

| Reagent type (species) or resource | Designation | Source or reference | Identifiers | Additional information |
|---|---|---|---|---|
| Strain, strain background *(E. coli)* | K-12 BW25113 Δ*tdcE* | Keio collection | Catalog # OEC4988 | |
| Strain, strain background *(E. coli)* | K-12 BW25113 Δ*hycB* | Keio collection | Catalog # OEC4988 | |
| Strain, strain background *(E. coli)* | K-12 BW25113 Δ*hycC* | Keio collection | Catalog # OEC4988 | |
| Strain, strain background *(E. coli)* | K-12 BW25113 Δ*hycD* | Keio collection | Catalog # OEC4988 | |
| Strain, strain background *(E. coli)* | K-12 BW25113 Δ*fdoH* | Keio collection | Catalog # OEC4988 | |
| Strain, strain background *(E. coli)* | K-12 BW25113 Δ*hybA* | Keio collection | Catalog # OEC4988 | |
| Strain, strain background *(E. coli)* | K-12 BW25113 Δ*nuoC* | Keio collection | Catalog # OEC4988 | |
| Strain, strain background *(E. coli)* | K-12 BW25113 Δ*atpG* | Keio collection | Catalog # OEC4988 | |
| Recombinant DNA reagent | *pMSs201 (kan$^R$)* | Dharmacon Promoter Library | Catalog # OEC4988 | |
| Recombinant DNA reagent | pUA66-EV (empty vector) | Gift from Dr. Mark P. Brynildsen | | A DNA fragment including *T5* promoter, *Kan$^R$* gene, pUA66 origin of replication and *lacI$^q$* was amplified from the pUA66-*gfp* plasmid with primers having BspHI cut sites. The amplified DNA fragment was digested with BspHI, and then self-ligated to obtain the modified pUA66-EV that does not have the *gfp* gene. |
| Recombinant DNA reagent | pUA66-*crp* | This study | | The *crp* gene with its promoter was amplified from the genomic DNA of *E. coli*, using forward and reverse primers with BglII and ScaI restriction enzyme cut sites, respectively. The pUA66-*gfp* plasmid was double digested with BglII and ScaI to remove *T5* promoter region and *gfp* gene. Then, the digested *crp* gene with its promoter and plasmid were ligated to generate pUA66-*crp*. |
| Strain, strain background *(E. coli)* | K-12 MG1655 pMSs201 P$_{sdhABCD}$-*gfp* | This study | | |
| Strain, strain background *(E. coli)* | K-12 MG1655 Δ*crp* pMSs201 P$_{sdhABCD}$-*gfp* | This study | | |
| Strain, strain background *(E. coli)* | K-12 MG1655 Δ*cyaA* pMSs201 P$_{sdhABCD}$-*gfp* | This study | | |
| Strain, strain background *(E. coli)* | K-12 MG1655 pMSs201 P$_{cyaA}$-*gfp* | This study | | |
| Strain, strain background *(E. coli)* | K-12 MG1655 Δ*crp* pMSs201 P$_{cyaA}$-*gfp* | This study | | |
| Strain, strain background *(E. coli)* | K-12 MG1655 Δ*cyaA* pMSs201 P$_{cyaA}$-*gfp* | This study | | |
| Strain, strain background *(E. coli)* | K-12 MG1655 Δ*crp* pUA66-EV | This study | | |
| Strain, strain background *(E. coli)* | K-12 MG1655 Δ*crp* pUA66-*crp* | This study | | |
| Software, algorithm | Prism (version 10.3.0) | GraphPad | RRID:SCR_002798 | http://www.graphpad.com/ |

*Appendix 1 Continued*

| Reagent type (species) or resource | Designation | Source or reference | Identifiers | Additional information |
|---|---|---|---|---|
| Software, algorithm | FlowJo (version 10.8.1) | Becton, Dickinson & Company | RRID:SCR_008520 | https://www.flowjo.com/ |
| Software, algorithm | MATLAB (version R2020b) | MathWorks | RRID:SCR_001622 | https://www.mathworks.com/ |
| Sequence-based reagent | Forward Primer (5' to 3') Δ*crp*::KAN(R) | This study | Integrated DNA Technologies, Inc. | TCTGGCTCTGGAGAAAGCT TATAACAGAGGATAACCGCGCG TGTAGGCTGGAGCTGCTTC |
| Sequence-based reagent | Reverse Primer (5' to 3') Δ*crp*::KAN(R) | This study | Integrated DNA Technologies, Inc. | AAAATGGCGCGCTACC AGGTAACGCGCCACTCC GACGGGATTAACGGCTGA CATGGGAAT |
| Sequence-based reagent | Forward Primer (5' to 3') Δ*cyaA*::KAN(R) | This study | Integrated DNA Technologies, Inc. | GAATCACAGTCATGACG GGTAGCAAATCAGGCGATA CGTCGTGTAGGCTGGAGCTGCTTC |
| Sequence-based reagent | Reverse Primer (5' to 3') Δ*cyaA*::KAN(R) | This study | Integrated DNA Technologies, Inc. | AGATTGCATGCCGGATA AGCCTCGCTTTCCGGCAC GTTCATTAACGGCTGACATGGGAAT |
| Sequence-based reagent | Forward Primer (5' to 3') Δ*sucA*::KAN(R) | This study | Integrated DNA Technologies, Inc. | ACGGCGAAGTAAGCATAA AAAAGATGCTTAAGGGATCA CGGTGTAGGCTGGAGCTGCTTC |
| Sequence-based reagent | Reverse Primer (5' to 3') Δ*sucA*::KAN(R) | This study | Integrated DNA Technologies, Inc. | GGTCAGGGACCAGAATAT CTACGCTACTCATTGTGTAT CCTTTATTTAACGGCTGACA TGGGAAT |
| Sequence-based reagent | Forward Primer (5' to 3') Δ*lpd*::KAN(R) | This study | Integrated DNA Technologies, Inc. | GACGGGTATGACCGCC GGAGATAAATATATAGAGG TCATGGTGTAGGCTGGAGCTGCTTC |
| Sequence-based reagent | Reverse Primer (5' to 3') Δ*lpd*::KAN(R) | This study | Integrated DNA Technologies, Inc. | GCCGCTTTTTTAATTGCC GGATGTTCCGGCAAACGAA AAATTAACGGCTGACATGGGAAT |
| Sequence-based reagent | Forward Primer (5' to 3') Δ*sucC*::KAN(R) | This study | Integrated DNA Technologies, Inc. | GGTTTAAAAGATAACGATT ACTGAAGGATGGACAGAACA CGTGTAGGCTGGAGCTGCTTC |
| Sequence-based reagent | Reverse Primer (5' to 3') Δ*sucC*::KAN(R) | This study | Integrated DNA Technologies, Inc. | TGGCAGATAACCTTGGTG TTTTTATCGATTAAAATGGAC ATTAACGGCTGACATGGGAAT |
| Sequence-based reagent | Forward Primer (5' to 3') Δ*sdhA*::KAN(R) | This study | Integrated DNA Technologies, Inc. | TTTACGTGATTTATG GATTCGTTGTGGTGT GGGGTGTGTGGTGTA GGCTGGAGCTGCTTC |
| Sequence-based reagent | Reverse Primer (5' to 3') Δ*sdhA*::KAN(R) | This study | Integrated DNA Technologies, Inc. | GATAAATTGAAAACT CGAGTCTCATTTTCC TGTCTCCGCATTAAC GGCTGACATGGGAAT |
| Sequence-based reagent | Forward Primer (5' to 3') Δ*gltA*::KAN(R) | This study | Integrated DNA Technologies, Inc. | TAAGTTCCGGCAGTCTT ACGCAATAAGGCGCTAAG GAGACCTTAAGTGTAGGCT GGAGCTGCTTC |
| Sequence-based reagent | Reverse Primer (5' to 3') Δ*gltA*::KAN(R) | This study | Integrated DNA Technologies, Inc. | CCCGCCATATGAACGGCG GGTTAAAATATTTACAACTTAGCA ATCAACCATTAACGGCTGACATGGGAAT |
| Sequence-based reagent | Forward Primer (5' to 3') Δ*aceE*::KAN(R) | This study | Integrated DNA Technologies, Inc. | GGTTCCAGAAAACTCAACGTTA TTAGATAGATAAGGAATAACCCGTGT AGGCTGGAGCTGCTTC |

*Appendix 1 Continued on next page*

*Appendix 1 Continued*

| Reagent type (species) or resource | Designation | Source or reference | Identifiers | Additional information |
|---|---|---|---|---|
| Sequence-based reagent | Reverse Primer (5′ to 3′) Δ*aceE*::KAN(R) | This study | Integrated DNA Technologies, Inc. | GCCCCGATGTCCGGTACTT TGATTTCGATAGCCATTATTCT TTTACCTCTTAACGGCTGACAT GGGAAT |
| Sequence-based reagent | Forward Primer (5′ to 3′) Δ*mdh*::KAN(R) | This study | Integrated DNA Technologies, Inc. | GCGGAGCAACATATCTTAG TTTATCAATATAATAAGGAGTTT AGGGTGTAGGCTGGAGCTGCTTC |
| Sequence-based reagent | Reverse Primer (5′ to 3′) Δ*mdh*::KAN(R) | This study | Integrated DNA Technologies, Inc. | CCGGAGTCTGTGCTCCGGT TTTTTATTATCCGCTAATCAATT AACGGCTGACATGGGAAT |
| Sequence-based reagent | Forward Primer (5′ to 3′) Δ*nuoI*::KAN(R) | This study | Integrated DNA Technologies, Inc. | CTGTCATTCTCTGGCAGG CGCAATAAGGGGCAATAAGA CCGTGTAGGCTGGAGCTGCTTC |
| Sequence-based reagent | Reverse Primer (5′ to 3′) Δ*nuoI*::KAN(R) | This study | Integrated DNA Technologies, Inc. | AGGCCACAGATATAAAAAGC GAACTCCATTGCCCCTCTCCTT AACGGCTGACATGGGAAT |
| Sequence-based reagent | Forward Primer (5′ to 3′) Δ*nuoM*::KAN(R) | This study | Integrated DNA Technologies, Inc. | TCCGGTCCTGACGGGACT TTTACAAGGAATAAAGATCGCC GTGTAGGCTGGAGCTGCTTC |
| Sequence-based reagent | Reverse Primer (5′ to 3′) Δ*nuoM*::KAN(R) | This study | Integrated DNA Technologies, Inc. | GCAGTGCGATCAGGTTTTG TGGAGTTATTGTCATGGCGAT TTAACGGCTGACATGGGAAT |
| Sequence-based reagent | Forward Primer (5′ to 3′) Δ*atpA*::KAN(R) | This study | Integrated DNA Technologies, Inc. | GCGCCTTGCAGACGTCTTGC AGTCTTAAGGGGACTGGAGCGT GTAGGCTGGAGCTGCTTC |
| Sequence-based reagent | Reverse Primer (5′ to 3′) Δ*atpA*::KAN(R) | This study | Integrated DNA Technologies, Inc. | TCAATGCCTTGCGGCCTGCCCT AAGGCAAGCCGCCAGACGTTAAC GGCTGACATGGGAAT |
| Sequence-based reagent | Forward Primer (5′ to 3′) Δ*atpB*::KAN(R) | This study | Integrated DNA Technologies, Inc. | TGGCACCGGCTGTAATTAA CAACAAAGGGTAAAAGGCATC GTGTAGGCTGGAGCTGCTTC |
| Sequence-based reagent | Reverse Primer (5′ to 3′) Δ*atpB*::KAN(R) | This study | Integrated DNA Technologies, Inc. | CTCCAGTTTGTTTCAGTTA AAACGTAGTAGTGTTGGTAAAT TAACGGCTGACATGGGAAT |
| Sequence-based reagent | Forward Primer (5′ to 3′) Δ*atpC*::KAN(R) | This study | Integrated DNA Technologies, Inc. | GGAAAAAGCCAAAAAACTTT AACGCCTTAATCGGAGGGTGATG TGTAGGCTGGAGCTGCTTC |
| Sequence-based reagent | Reverse Primer (5′ to 3′) Δ*atpC*::KAN(R) | This study | Integrated DNA Technologies, Inc. | GCCTGTTTCCAGACTGGCTTT TGTGCTTTTCAAGCCGGTGTTAA CGGCTGACATGGGAAT |
| Sequence-based reagent | Forward Primer (5′ to 3′) Δ*atpD*::KAN(R) | This study | Integrated DNA Technologies, Inc. | CCGCCGCGGTTTAAACAGGTT ATTTCGTAGAGGATTTAAGGTGTA GGCTGGAGCTGCTTC |
| Sequence-based reagent | Reverse Primer (5′ to 3′) Δ*atpD*::KAN(R) | This study | Integrated DNA Technologies, Inc. | AGGTGGTAAGTCATTGCCATAT CACCCTCCGATTAAGGCGTTAAC GGCTGACATGGGAAT |
| Sequence-based reagent | Forward Primer (5′ to 3′) Δ*frdC*::KAN(R) | This study | Integrated DNA Technologies, Inc. | TTTCTTATCGCGACCCTGAAA CCACGCTAAGGAGTGCAACGTG TAGGCTGGAGCTGCTTC |
| Sequence-based reagent | Reverse Primer (5′ to 3′) Δ*frdC*::KAN(R) | This study | Integrated DNA Technologies, Inc. | GTCAGAACGCTTTGGATTTGG ATTAATCATCTCAGGCTCCTTAAC GGCTGACATGGGAAT |

*Appendix 1 Continued on next page*

*Appendix 1 Continued*

| Reagent type (species) or resource | Designation | Source or reference | Identifiers | Additional information |
|---|---|---|---|---|
| Sequence-based reagent | Forward Primer (5′ to 3′) Δ*pgi*::KAN(R) | This study | Integrated DNA Technologies, Inc. | GCTACAATCTTCCAAAGTCACA ATTCTCAAAATCAGAAGAGTATTGC TAGTGTAGGCTGGAGCTGCTTC |
| Sequence-based reagent | Reverse Primer (5′ to 3′) Δ*pgi*::KAN(R) | This study | Integrated DNA Technologies, Inc. | GCGGCGTGAACGCCTTATCC GGCCTACATATCGACGATGATTA ACGGCTGACATGGGAAT |
| Sequence-based reagent | Forward Primer (5′ to 3′) Δ*zwf*::KAN(R) | This study | Integrated DNA Technologies, Inc. | CTGGCTTAAGTACCGGGTTAG TTAACTTAAGGAGAATGACGTGTA GGCTGGAGCTGCTTC |
| Sequence-based reagent | Reverse Primer (5′ to 3′) Δ*zwf*::KAN(R) | This study | Integrated DNA Technologies, Inc. | GCGCAAGATCATGTTACC GGTAAAATAACCATAAAGGA TAAGCGCAGATATTAACGGC TGACATGGGAAT |
| Sequence-based reagent | Forward Primer (5′ to 3′) Δ*tktB*::KAN(R) | This study | Integrated DNA Technologies, Inc. | CTTCTTGCCGCCAAACT ATAAACCAGCCACGGAGTG TTATGTGTAGGCTGGAGCTG CTTC |
| Sequence-based reagent | Reverse Primer (5′ to 3′) Δ*tktB*::KAN(R) | This study | Integrated DNA Technologies, Inc. | GTCAGCGTCGCATCCGGCAA TCAGCATCCGGCAATCACCATTA ACGGCTGACATGGGAAT |
| Sequence-based reagent | Forward Primer (5′ to 3′) Δ*talA*::KAN(R) | This study | Integrated DNA Technologies, Inc. | CGCACTCATCTAACACTTTACT TTTCAAGGAGTATTTCCTGTGTAGG CTGGAGCTGCTTC |
| Sequence-based reagent | Reverse Primer (5′ to 3′) Δ*talA*::KAN(R) | This study | Integrated DNA Technologies, Inc. | GGCAAGGTCTTTTCGGGAC ATATAACACTCCGTGGCTGGT TTAACGGCTGACATGGGAAT |
| Sequence-based reagent | Forward Primer (5′ to 3′) pUA66-*crp* | This study | Integrated DNA Technologies, Inc. | GCGCTCAGATCTTGATC CGAAAGCTATGCTAAAACAGT |
| Sequence-based reagent | Reverse Primer (5′ to 3′) pUA66-*crp* | This study | Integrated DNA Technologies, Inc. | GCGCTCAGTACTttaAC GAGTGCCGTAAACGA |

