## [Editor Report · eLife Assessment]

The study reports an **important** finding on the role of the global metabolic regulator Crp/cAMP in the formation of antibiotic persister *Escherichia coli*. The evidence supporting the claims is **solid** including metabolomic analysis and characterization of many mutant strains.

---

## [Referee Report · Reviewer #1 (Public review)]

The authors set out to understand the role played by a key global metabolic regulator called Crp/cAMP in the formation of persister *Escherichia coli* that survive antibiotic treatment without acquiring genetic mutations.

In order to achieve this aim, the authors employ an interdisciplinary approach integrating standard microbiology assays with cutting-edge genomic, metabolomic and proteomics screening.

The data presented by the authors convincingly demonstrate that the deletion of two key genes that are part of the Crp/cAMP complex (i.e. crp and cyaA) leads to a significant decrease in the number of *E. coli*.

The authors have carried out additional experiments to further validate this point by using the well characterised hipA7 *E. coli* mutant.

The data presented also demonstrate that deletion of the crp gene leads to an overall decrease in energy metabolism and an overall increase in anabolic metabolism at the population level. The deletion of cyaA has an opposite effect on cAMP concentration compared to crp deletion, the authors presented a possible hypotheses but did not test it.

The authors have now explicitly acknowledged in their discussion that the data presented in this study are obtained at the whole population level rather than at the level of the persister subpopulation and therefore should be considered with caution.

Finally, the authors convincingly show that the persisters they investigated are non-growing and have a higher redox activity and that the deletion of key genes involved in energy metabolism leads to a decrease in the number of persisters.

These data will be important for future investigations on the biochemical mechanisms that allow bacteria to adapt to stressors such as nutrient depletion or exposure to antibiotics. As such this work will likely have an impact in a variety of fields such as bacterial biochemistry, antimicrobial resistance research and environmental microbiology.

Strengths:

Interdisciplinary approach.

Excellent use of replication and ensuring reproducibility.

Excellent understanding and presentation of the biochemical mechanisms underpinning bacterial physiology via an integrated genomic, metabolomic and proteomic screening.

Weaknesses:

There is no tested mechanisms explaining why the deletion of cyaA has an opposite effect on cAMP concentration compared to crp deletion.

Metabolomics, proteomics and metabolic activity data are obtained at the whole population level rather than at the level of the persister sub-population.

---

## [Author Response]

The following is the authors’ response to the original reviews.

**Reviewer #1:**
(1) Two genes from the Crp/cAMP complex (*crp* and *cyaA*) are hypothesized to be key for persistence but key metabolomics and proteomics data are obtained from only one deletion mutant in the crp gene.

We thank the reviewer for their thoughtful assessment of our manuscript and for providing valuable comments.

In our study, we have demonstrated that deletion of both *cyaA* and crp genes results in the same persistence phenotype. In a previous study, we screened knockout strains of global transcriptional regulators using the aminoglycoside (AG) potentiation assay and found that, across a panel of carbon sources, AG potentiation occurred in tolerant cells derived from most knockout strains—except for Δ*crp* and Δ*crp* (Mok et al., 2015). This indicated that both genes are critical components of the Crp/cAMP regulatory network in persistence. Because cAMP exerts its effects when bound to its receptor protein Crp, disrupting crp alone should effectively abolish Crp/cAMP complex function (Keseler et al., 2011). Thus, we reasoned that comparing Δ*crp* to wild-type would be sufficient to capture the key metabolic and proteomic alterations arising from Crp/cAMP perturbation. Given the substantial cost and labor intensity of untargeted metabolomics and proteomics analyses, this experimental design allowed us to extract meaningful insights while maintaining feasibility. Nonetheless, to ensure the robustness of our findings, we have conducted all subsequent validation experiments using both Δ*crp* and Δ*crp* strains, confirming that the observed metabolic and proteomic changes are consistent across both mutants. We have now provided a concise justification statement in the manuscript (see lines 197-200 in the current manuscript).

(2) The deletion of crp and *crp* have opposite effects on the concentration of cAMP, a comparison of metabolomics and proteomics data obtained using both mutants might aid in understanding this difference.

Although this is an interesting outcome, we have already discussed in the manuscript that it is likely due to the feedback regulation of the Crp/cAMP complex on *crp* expression (see Fig. 1 Keseler et al., 2011) (Aiba, 1985; Keseler et al., 2011; Majerfeld et al., 1981). Specifically, perturbation of the Crp/cAMP complex by deleting *crp* should enhance *crp* promoter (P*crp*) activity, leading to increased CyaA protein expression and, consequently, elevated intracellular cAMP levels. To experimentally verify this predicted feedback regulation, we utilized *E. coli* K-12 MG1655 WT, Δ*crp*, and Δ*crp* strains harboring the pMSs201 plasmid, which encodes green fluorescent protein (*gfp*) under the control of the P*_cyaA_* promoter. This design allowed us to directly assess the effect of Crp/cAMP perturbation on P*_cyaA_* activity by quantifying *gfp* expression as a reporter. By comparing the mutant strains to WT, we could determine whether loss of Crp/cAMP function indeed derepresses *crp* expression. As expected, genetic perturbation of Crp/cAMP enhanced P*_cyaA_* promoter activity, resulting in increased gfp expression (Figure 1-figure supplement 2). This result supports the role of Crp/cAMP in regulating *crp* expression via feedback control. We have now explicitly discussed this rationale in the manuscript and included the corresponding data (see lines 410-418 and Figure 1-figure supplement 2 in the current manuscript).

(3) Metabolomics, proteomics, and metabolic activity data are obtained at the whole population level rather than at the level of the persister sub-population.

Performing metabolomic, proteomic, and other assays at the level of the persister subpopulation is inherently challenging in this study and across the persister research field, as it requires isolating a pure persister population. While metabolic inhibitors like rifampin and tetracycline can induce dormancy and antibiotic tolerance in the entire population (Kwan et al., 2013), these treatments generate artificially altered cell states that may not accurately reflect naturally occurring persisters. Fluorescent reporters combined with fluorescence-activated cell sorting (FACS) have been utilized to study persister cells, including in our previous studies (Amato et al., 2013; Orman & Brynildsen, 2013, 2015). However, this approach only enriches for persisters rather than isolating a pure population, as persisters still constitute a small fraction of the sorted cells (Amato et al., 2013; Orman & Brynildsen, 2013, 2015). Despite these limitations, our untargeted metabolomics and proteomics analyses at the whole-population level provide valuable insights into the regulatory mechanisms of the Crp/cAMP complex and its potential role in persister formation. We have rigorously examined the impact of these mechanisms on non-growing cell formation (see Figure 4 in the current manuscript) and persister levels (see Figure 5 in the current manuscript) through flow cytometry and single-gene deletion experiments. We appreciate the reviewer’s comment and have acknowledged and discussed these methodological challenges in our manuscript (see lines 397-406 in the current manuscript).

**Reviewer #2:**
(1) The approaches used here are aimed at the major bacterial population, but yet the authors used the data reflecting the major population behavior to interpret the physiology of persister cells that comprise less than 1% of the major bacterial population. How they can pick up a needle from the hay without being fooled by the spill-over artifacts from the major population? Although it is probably very difficult to isolate and directly assay persister cells, firm conclusions for the type proposed by the authors cannot be firmly established without such assays. Perhaps introducing *crp*/crp mutation into the best example of persistence, the hipA-7 high persistence phenotype may clarify this issue to a certain extent.

We thank the reviewer for their thoughtful assessment of our manuscript and for providing valuable comments.

Performing metabolomics and proteomics at the level of the persister subpopulation remains a major challenge in this study and across the persister research field, as it requires isolating a pure persister population. While metabolic inhibitors like rifampin and tetracycline can induce dormancy and antibiotic tolerance in the entire population (Kwan et al., 2013), these treatments generate artificially altered cell states that may not accurately reflect naturally occurring persisters. Similarly, fluorescent reporters combined with fluorescence-activated cell sorting (FACS) have been employed to study persister cells, including in our previous studies (Amato et al., 2013; Orman & Brynildsen, 2013, 2015). However, this approach only results in persister-enriched populations rather than a pure isolate, meaning that persisters still constitute a small fraction of the sorted cells (Amato et al., 2013; Orman & Brynildsen, 2013, 2015). Despite these inherent limitations, our untargeted metabolomics and proteomics analyses at the whole-population level provide valuable insights into the regulatory mechanisms of the Crp/cAMP complex and its potential role in persister formation. Specifically, our data reveal clear indications that Crp/cAMP activity promotes the formation of a non-growing cell subpopulation, while its deletion reduces this effect. We have validated this observation through single-cell analyses (see Figure 4 in the current manuscript). Additionally, our data strongly suggest that energy metabolism plays a critical role in persister cell physiology, and we have rigorously tested this hypothesis using persister assays for single-gene deletions (see Figure 5 in the current manuscript).

Furthermore, in response to the reviewer’s suggestion, we introduced *crp* and *crp* deletions into the HipA-7 high-persistence mutant strain. The impact of these deletions in HipA-7 mirrored their effects in the wild-type strain (Figure 1-figure supplement 8), further supporting our conclusions. This data has been provided and discussed in the manuscript (see lines 185-189, and Figure 1-figure supplement 8 in the current manuscript).

We acknowledge the challenges in directly assaying persister cells, and we have now discussed this in the manuscript (see lines 397-406 in the current manuscript).

(2) The authors overlooked/omitted a recently published work regarding *cyaA* and crp (PMID: 35648826). In that work, a deficiency in *cyaA* or crp confers tolerance to diverse types of lethal stressors, including all lethal antimicrobials tested. How a mutation conferring pan-tolerance to the major bacterial population would lead to a less protective effect with a minor subpopulation? The authors are kind of obligated to discuss such a paradox in the context of their work because that is the most relevant literature for the present work. It is also very interesting if the *cyaA*/crp deficiency really has an opposing effect on tolerance and persistence. As a note, most of the conclusions from the omics studies of the present work have been reached in that overlooked literature, which addresses mechanisms of tolerance, a major rather than a minor population behavior. That supports comment #1 above. The inability of the authors to observe tolerance phenotype with the *cyaA* or crp mutant possibly derived from extremely high antimicrobial concentrations used in the study prevents tolerance phenotype from being observed because tolerance is sensitive to antimicrobial concentration while persistence is not.(3) The authors overly stressed the effect of *cyaA*/crp on persister formation but failed to test an alternative explanation of their effect on persister waking up after antimicrobial treatment. If the *cyaA*/crp-derived persisters are put into deeper sleep during antimicrobial treatment than wildtype-derived persisters, a 16-h recovery growth might have underestimated viable bacteria. This is often the case especially when extremely high concentrations of antimicrobials are used in performing persister assay. Thus, at least a longer incubation time (e.g. 48 and 72h) of agar plates for persister viable count needs to be performed to test such a scenario.(4) The rationale for using extremely high drug concentrations to perform persister assay is unclear. There are 2 issues with using extremely high drug concentrations. First, when overly high concentrations are used, drug removal becomes difficult. For example, a two-time wash will not be able to bring drug concentration from > 100 x MIC to below MIC. This is especially problematic with aminoglycoside because drug removal by washing does not work well with this class of compound. Second, overly high concentrations of drug use may make killing so rapidly and severely that may mask the difference from being observed between mutants and the control wild-type strain. In such cases, you would need to kill over a wide range of drug concentrations to find the right window to show a difference. The gentamicin data in the present work is likely the case that needs to be carefully examined. The mutants and the wild-type strain have very different MICs for gentamicin, but a single absolute drug concentration rather than concentrations normalized to MIC was used. This is like to compare a 12-year-old with a 21-year-old to run a 100-meter dash, which is highly inappropriate.

The reviewer notes that key literature (PMID: 35648826) was overlooked, showing *cyaA*/crp deficiency confers broad stress tolerance—contradicting the reported reduction in persister protection. They suggest high drug concentrations may mask tolerance, and also, longer incubation (48–72 h) and normalized drug levels based on MIC are recommended. Given that these three independent comments are interconnected, we will address them together.

We follow a rigorous washing protocol to minimize antibiotic carryover. After treatment, 1 ml of culture is centrifuged at 13,300 RPM (17,000 x g) for 3 minutes, and >950 µl of supernatant is removed without disturbing the pellet. The pellet is resuspended in 950 µl PBS, diluting antibiotics >20-fold. This step is repeated, resulting in a >400-fold cumulative dilution. After the final wash, cells are resuspended in 100 µl PBS, then serially diluted and plated on antibiotic-free agar to ensure consistency and eliminate residual antibiotics. Preliminary experiments are routinely done in our laboratory to confirm the effectiveness of washing procedures. To address concerns that high antibiotic concentrations may mask phenotypic differences—particularly in the gentamicin assay—we conducted additional experiments using MIC-normalized doses (5×, 10×, and the original study concentration) with six wash steps. As shown in Figure 1-figure supplement 6, all concentrations consistently reduced persister levels, supporting our original findings. While 5× MIC ampicillin allowed detection of persisters in mutant strains, their levels remained multiple orders of magnitude lower than in wild-type, maintaining statistical significance. These results, along with updated washing protocols, are now included in the revised manuscript (see lines 176-185 and Figure 1-figure supplement 6 in the current manuscript).

Although we standardize the incubation time of the agar plates for all conditions and strains, most strains form sufficiently large colonies within 16 hours, and longer incubation often leads to large, overlapping colonies that hinder accurate counting. We assure the reviewer that we always leave the plates in the incubator beyond the initial counting period to monitor the emergence of any new colonies. Here, we provide plate images of key strains after antibiotic treatments, demonstrating that extended incubation did not alter CFU levels, as shown in Figure 1-figure supplement 7. We have updated the relevant section in the Materials and Methods to clarify this point and included the plate images in the current manuscript (see lines 181-182 and Figure 1-figure supplement 7 in the current manuscript).

We acknowledge the significance of the study highlighted by the reviewer (Zeng et al., 2022); however, direct comparisons with our results are challenging due to substantial differences in experimental conditions, antibiotic concentrations, treatment durations, and most importantly, the *E. coli* strains used. The study of Zeng et al., 2022, utilized strains from the Keio collection, a commercially available *E. coli* BW25113 mutant library, which may contain unknown background mutations that could influence tolerance phenotypes. While we used the Keio collection for initial screening, we always validate single clean deletions in our lab strain, *E. coli* MG1655, to ensure robust conclusions. The observed variations in tolerance and persistence between studies can largely be attributed to these methodological differences rather than an inherent paradox. The concentrations of ampicillin (200 µg/mL) and ofloxacin (5 µg/mL) used in our assays are in line with concentrations employed in foundational persister studies (Amato & Brynildsen, 2015; Cui et al., 2016; Hansen et al., 2008; Leszczynska et al., 2013; Lin et al., 2022; Orman & Brynildsen, 2015; Shah et al., 2006). These levels represent >10 × the MIC and are necessary to ensure the elimination of actively growing cells, thus enriching for persister cells that, by definition, survive high bactericidal drug exposure. Our aim is not to model pharmacokinetics per se, but to apply a standardized challenge to distinguish phenotypic persistence. Furthermore, pharmacokinetic and pharmacodynamic clinical data show that antibiotics such as ofloxacin and ampicillin can reach levels far exceeding 10× MIC for extended periods in patients (OFLOXACIN, 2019; Soto et al., 2014).

To assess how *cyaA* and crp deletions affect antibiotic responses under conditions similar to those used by Zeng et al. (Zeng et al., 2022) —specifically, exponential-phase *E. coli* BW25113 strains (Keio collection), lower antibiotic concentrations, and short treatments (e.g., 1 hour)—we first tested *E. coli* MG1655 WT, Δ*crp*, and Δ*crp* strains in late stationary phase using reduced antibiotic concentrations and shorter exposures. Both knockouts showed decreased survival following ampicillin and ofloxacin treatment compared to WT (see Figure 1-figure supplement 6), consistent with our findings in Figure 1 in the manuscript. In exponential phase, the knockout strains exhibited reduced survival after ampicillin treatment but increased survival after ofloxacin treatment relative to WT (see Author response image 2A below), again mirroring the trends in Figure 1. Gentamicin treatment, however, produced variable results in MG1655 knockouts, likely due to the brief 1-hour exposure being insufficient for robust conclusions (Author response image 2A). Notably, when we tested the corresponding Keio knockout strains in the BW25113 background, we observed increased tolerance in exponential-phase cells, reproducing Zeng et al.'s findings under their specific conditions (see Author response image 2B below), although BW25113 and MG1655 exhibited distinct persister phenotypes in exponential phase (Author response image 2A, B). These results, altogether, highlight the sensitivity of antibiotic tolerance and persistence phenotypes to factors such as strain background, antibiotic concentration, and treatment duration. This is now discussed in detail in the revised manuscript, with supporting data provided (see lines 460-476, and Supplement File 6, 7 in the current manuscript).

**Author response image 1. sa2fig1:** Persister levels of *E. coli* K-12 MG1655 WT, Δ*crp*, and Δ*crp* strains in late stationary phase. Cells were treated with ampicillin (5× MIC for 4 h), ofloxacin (5× MIC for 2.5 h), and gentamicin (3× MIC for 1 h). Concentrations and treatment durations were selected based on (Zeng et al., 2022).

**Author response image 2. sa2fig2:** Persister levels of *E. coli* K-12 MG1655 (Panel A) and BW25113 (Panel B) WT, Δ*crp*, and Δ*crp* strains in the exponential growth phase. Cells were treated at mid-exponential phase (OD_600_ ~0.25) with ampicillin (5× MIC for 4 h), ofloxacin (5× MIC for 2.5 h), and gentamicin (3× MIC for 1 h). Treatment concentrations and durations were based on conditions described in (Zeng et al., 2022).

**Reviewer #3:**
The authors try to draw too many conclusions and it's difficult to identify what their actual findings are. For instance, they do not have any interesting findings with aminoglycosides but include the data and spend a lot of time discussing it, but it is really a distraction. The correlation between the induction of anabolic pathways in the crp mutant in the late stationary phase and the reduction in persisters is potentially very interesting but is buried in the paper with the vast quantities of data, and observations and conclusions that are often not well substantiated.

We thank the reviewer for their assessment that helped us clarify and strengthen the focus of our manuscript.

While our study is not focused on aminoglycosides, we believe the related data provide important insights into persister cell physiology. Persisters are traditionally described as metabolically dormant, non-growing cells. However, we consistently observe that aminoglycosides—despite requiring energy-dependent uptake and active protein translation for their activity—can still eliminate persister cells in wild-type *E. coli*. This finding supports our central hypothesis that persisters may retain a basal level of metabolic activity sufficient to permit aminoglycoside uptake and action during prolonged treatment. We have revised the manuscript to present this point more clearly, ensuring it complements rather than distracts from the main narrative.

We respectfully emphasize that our conclusions are supported by multiple layers of evidence. Our metabolomics data are corroborated by proteomics and further validated by functional assays, including redox state measurements, growing versus non-growing cell detection, and targeted persister assays. In addition, we performed labor-intensive validations using individually selected Keio mutants treated with antibiotics to quantify persister levels, with key observations further confirmed in single-gene deletions in *E. coli* MG1655 strains.

We believe the revisions made in response to all reviewers’ comments have significantly improved the clarity, focus, and overall impact of the manuscript.

The discussion section is particularly difficult to read and I recommend a large overhaul to increase clarity. For instance, what are the authors trying to conclude in section (iii) of the discussion? That persisters in the stationary phase have higher energy than other cells? Is there data to support that? All sections are similarly lacking in clarity.

We repeatedly emphasize in the manuscript that while persister survival depends on energy metabolism, this does not imply that persisters have higher metabolic activity than those in the exponential growth phase. We have clarified this point in the revised manuscript (see lines 67-79, and 442-444 in the current manuscript).

The large number of mutants characterized is a strength, but the quality of the data provided for those experiments is poor. Did some of these mutants lose fitness in the deep stationary phase in the absence of antibiotics? Did some reach a far lower cfu/ml in the stationary phase? These details are important and without them, it is difficult to interpret the data.

Although metabolic mutations can affect cell growth, we do not observe substantial differences in cell numbers during the late stationary phase, when persister assays are performed. These knockout strains reach stationary phase fully by that time. We emphasize that we routinely measure cell numbers at this stage using flow cytometry before diluting cultures into fresh media and applying antibiotic treatments. Cell counts for the metabolic mutants are shown in Figure 5-figure supplement 4 in the current manuscript, and no significant growth deficiencies are observed in the late stationary phase. This is consistent with our previous publication (Shiraliyev & Orman, 2023) and findings from Lewis’s group (Manuse et al., 2021), where similar knockout strains showed no drastic impact on growth.

There is an analysis of persister formation in mutants in the pts/CRP pathway that is not discussed (Zeng et al PNAS 2022, Parsons et al PNAS, 2024).

These studies are now cited and discussed in the revised manuscript (see lines 459-476).

The authors do not discuss ROS production and antibiotic killing in these experiments. Presumably, the WT would have a greater propensity to produce ROS in response to antibiotics than the crp mutant, but it survives better. Is ROS not involved in antibiotic killing in these conditions?

The experimental conditions used here are identical to those in our previously published study on persister cells in the late stationary phase (Orman & Brynildsen, 2015), where we specifically investigated the role of ROS in antibiotic tolerance. In that work, we overexpressed key antioxidant enzymes—catalases (*katE*, *katG*) and superoxide dismutases (*sodA*, *sodB* and *sodC*)—at stationary phase. These enzymes were confirmed to be catalytically active through functional assays, yet their overexpression had no measurable effect on persister levels. To further decouple ROS from respiratory activity in that study, we performed anaerobic experiments using nitrate as an alternative terminal electron acceptor. We found that anaerobic respiration actually enhanced persister formation, and inhibition of nitrate reductases using KCN reduced it—again, independent of ROS. These findings provide compelling evidence that it is the respiratory activity itself, rather than ROS production, that influences persister formation in our system.

We have now included this discussion in the revised manuscript to clarify that ROS are unlikely to be a major factor in antibiotic killing under these conditions (see lines 503-513).

ReferencesAiba, H. (1985). Transcription of the *Escherichia coli* adenylate cyclase gene is negatively regulated by cAMP-cAMP receptor protein. The Journal of Biological Chemistry, 260(5), 3063–3070.

Amato, S. M., & Brynildsen, M. P. (2015). Persister Heterogeneity Arising from a Single Metabolic Stress. Current Biology, 25(16), 2090–2098. https://doi.org/10.1016/j.cub.2015.06.034

Amato, S. M., Orman, M. A., & Brynildsen, M. P. (2013). Metabolic Control of Persister Formation in *Escherichia coli*. Molecular Cell, 50(4), 475–487. https://doi.org/10.1016/J.MOLCEL.2013.04.002

Cui, P., Niu, H., Shi, W., Zhang, S., Zhang, H., Margolick, J., Zhang, W., & Zhang, Y. (2016). Disruption of Membrane by Colistin Kills Uropathogenic *Escherichia coli* Persisters and Enhances Killing of Other Antibiotics. Antimicrobial Agents and Chemotherapy, 60(11), 6867–6871. https://doi.org/10.1128/AAC.01481-16

Hansen, S., Lewis, K., & Vulić, M. (2008). Role of Global Regulators and Nucleotide Metabolism in Antibiotic Tolerance in *Escherichia coli*. Antimicrobial Agents and Chemotherapy, 52(8), 2718–2726. https://doi.org/10.1128/AAC.00144-08

Keseler, I. M., Collado-Vides, J., Santos-Zavaleta, A., Peralta-Gil, M., Gama-Castro, S., Muniz-Rascado, L., Bonavides-Martinez, C., Paley, S., Krummenacker, M., Altman, T., Kaipa, P., Spaulding, A., Pacheco, J., Latendresse, M., Fulcher, C., Sarker, M., Shearer, A. G., Mackie, A., Paulsen, I., … Karp, P. D. (2011). EcoCyc: a comprehensive database of *Escherichia coli* biology. Nucleic Acids Research, 39(Database), D583–D590. https://doi.org/10.1093/nar/gkq1143

Kwan, B. W., Valenta, J. A., Benedik, M. J., & Wood, T. K. (2013). Arrested protein synthesis increases persister-like cell formation. Antimicrobial Agents and Chemotherapy, 57(3), 1468–1473. https://doi.org/10.1128/AAC.02135-12

Leszczynska, D., Matuszewska, E., Kuczynska-Wisnik, D., Furmanek-Blaszk, B., & Laskowska, E. (2013). The Formation of Persister Cells in Stationary-Phase Cultures of *Escherichia coli* Is Associated with the Aggregation of Endogenous Proteins. PLoS ONE, 8(1), e54737. https://doi.org/10.1371/journal.pone.0054737

Lin, J. S., Bekale, L. A., Molchanova, N., Nielsen, J. E., Wright, M., Bacacao, B., Diamond, G., Jenssen, H., Santa Maria, P. L., & Barron, A. E. (2022). Anti-persister and Anti-biofilm Activity of Self-Assembled Antimicrobial Peptoid Ellipsoidal Micelles. ACS Infectious Diseases, 8(9), 1823–1830. https://doi.org/10.1021/acsinfecdis.2c00288

Majerfeld, I. H., Miller, D., Spitz, E., & Rickenberg, H. V. (1981). Regulation of the synthesis of adenylate cyclase in *Escherichia coli* by the cAMP — cAMP receptor protein complex. Molecular and General Genetics MGG, 181(4), 470–475. https://doi.org/10.1007/BF00428738

Manuse, S., Shan, Y., Canas-Duarte, S. J., Bakshi, S., Sun, W.-S., Mori, H., Paulsson, J., & Lewis, K. (2021). Bacterial persisters are a stochastically formed subpopulation of low-energy cells. PLoS Biology, 19(4), e3001194.

Mok, W. W. K., Orman, M. A., & Brynildsen, M. P. (2015). Impacts of global transcriptional regulators on persister metabolism. Antimicrobial Agents and Chemotherapy, 59(5), 2713–2719.

OFLOXACIN. (2019). https://dailymed.nlm.nih.gov/dailymed/fda/fdaDrugXsl.cfm?setid=1779c568-d7bb-4bd5-bc29-13bd52ba8a0a&type=display

Orman, M. A., & Brynildsen, M. P. (2013). Dormancy is not necessary or sufficient for bacterial persistence. Antimicrobial Agents and Chemotherapy, 57(7), 3230–3239.

Orman, M. A., & Brynildsen, M. P. (2015). Inhibition of stationary phase respiration impairs persister formation in *E. coli*. Nature Communications, 6(1), 7983.

Shah, D., Zhang, Z., Khodursky, A. B., Kaldalu, N., Kurg, K., & Lewis, K. (2006). Persisters: a distinct physiological state of *E. coli*. BMC Microbiology, 6(1), 53. https://doi.org/10.1186/1471-2180-6-53

Shiraliyev, R. C., & Orman, M. (2023). Metabolic disruption impairs ribosomal protein levels, resulting in enhanced aminoglycoside tolerance. BioRxiv, 2012–2023.

Soto, E., Shoji, S., Muto, C., Tomono, Y., & Marshall, S. (2014). Population pharmacokinetics of ampicillin and sulbactam in patients with community-acquired pneumonia: evaluation of the impact of renal impairment. British Journal of Clinical Pharmacology, 77(3), 509–521. https://doi.org/10.1111/bcp.12232

Zeng, J., Hong, Y., Zhao, N., Liu, Q., Zhu, W., Xiao, L., Wang, W., Chen, M., Hong, S., Wu, L., Xue, Y., Wang, D., Niu, J., Drlica, K., & Zhao, X. (2022). A broadly applicable, stress-mediated bacterial death pathway regulated by the phosphotransferase system (PTS) and the cAMP-Crp cascade. Proceedings of the National Academy of Sciences, 119(23). https://doi.org/10.1073/pnas.2118566119